# Substrate-interacting pore loops of two ATPase subunits determine the degradation efficiency of the 26S proteasome

Erika López-Alfonzo[1], Ayush Saurabh[2], Sahar Zarafshan[1], Connor Arkinson[1,3,4], Christine L. Gee[1,3,4], Hao-Hsuan Hsieh[1,3,4], Steve Pressé[2,5] & Andreas Martin[1,3,4] ✉

The 26S proteasome is the major eukaryotic protease responsible for the degradation of misfolded, damaged, and obsolete regulatory proteins. Commitment to degradation occurs when conserved pore loops in the hetero-hexameric ATPase motor of the proteasome engage the flexible initiation region of a polyubiquitinated protein substrate for subsequent mechanical unfolding and translocation into a proteolytic chamber. Here, we use in vitro biochemical assays, single-molecule FRET-based measurements, and cryo-EM structure determination to characterize how the pore-1 loops of individual ATPase subunits in the yeast 26S proteasome contribute to the different steps of substrate degradation and affect the proteasome conformational dynamics. We find that the pore-1 loops of the Rpt6 and Rpt4 ATPase subunits play particularly important, yet distinct roles in substrate capture and unfolding, and in holding the ATPase motor in a static state prior to substrate engagement. Interestingly, these pore-1 loop contributions correlate with the positions of ATPase subunits in spiral-staircase arrangements for the substrate-free and substrate-degrading proteasome, providing insights into the mechanisms of substrate processing by the 26S proteasome and related ATPase motors.

Protein degradation is essential for cellular homeostasis, cell division, differentiation, quality control, and the regulation of numerous other vital processes. The ubiquitin proteasome system (UPS) acts as the major pathway for energy-dependent protein degradation in eukaryotic cells, with the 26S proteasome as the final component recognizing and degrading targeted protein substrates[1]. This 2.5 MDa macromolecular machine is composed of the cylindrical 20S core particle (CP) and the 19S regulatory particle (RP) that caps the 20S CP on one or both ends (Fig. 1A). The 20S CP contains proteolytic active sites in an internal chamber for polypeptide cleavage, whereas the 19S RP is responsible for the recognition of ubiquitinated substrates, their deubiquitination, mechanical unfolding, and translocation into the proteolytic chamber. The 19S RP consists of the lid and base subcomplexes. The lid subcomplex with its nine subunits binds to the top and side of the base and core, and contains the essential $Zn^{2+}$-dependent deubiquitinase Rpn11[2-4]. The base is made up of 10 subunits, including three main ubiquitin receptors (Rpn1, Rpn10, and Rpn13)[5-8], the scaffolding subunit Rpn2, and six distinct ATPases (Rpt1–Rpt6) of the AAA+ (ATPases Associated with diverse cellular Activities) family that form the heterohexameric motor of the proteasome. These ATPase subunits, arranged in the order Rpt1, Rpt2, Rpt6, Rpt3, Rpt4, and Rpt5 (as seen from the top of the proteasome)[9], share a common domain architecture that includes a N-terminal coiled-coil, a small domain with an oligonucleotide/oligosaccharide-binding (OB) fold

[1]Department of Molecular & Cell Biology, University of California at Berkeley, Berkeley, CA, USA. [2]Department of Physics, Center for Biological Physics, Arizona State University, Tempe, AZ, USA. [3]California Institute for Quantitative Biosciences, University of California at Berkeley, Berkeley, CA, USA. [4]Howard Hughes Medical Institute, University of California at Berkeley, Berkeley, CA, USA. [5]School of Molecular Science, Arizona State University, Tempe, AZ, USA. ✉e-mail: a.martin@berkeley.edu

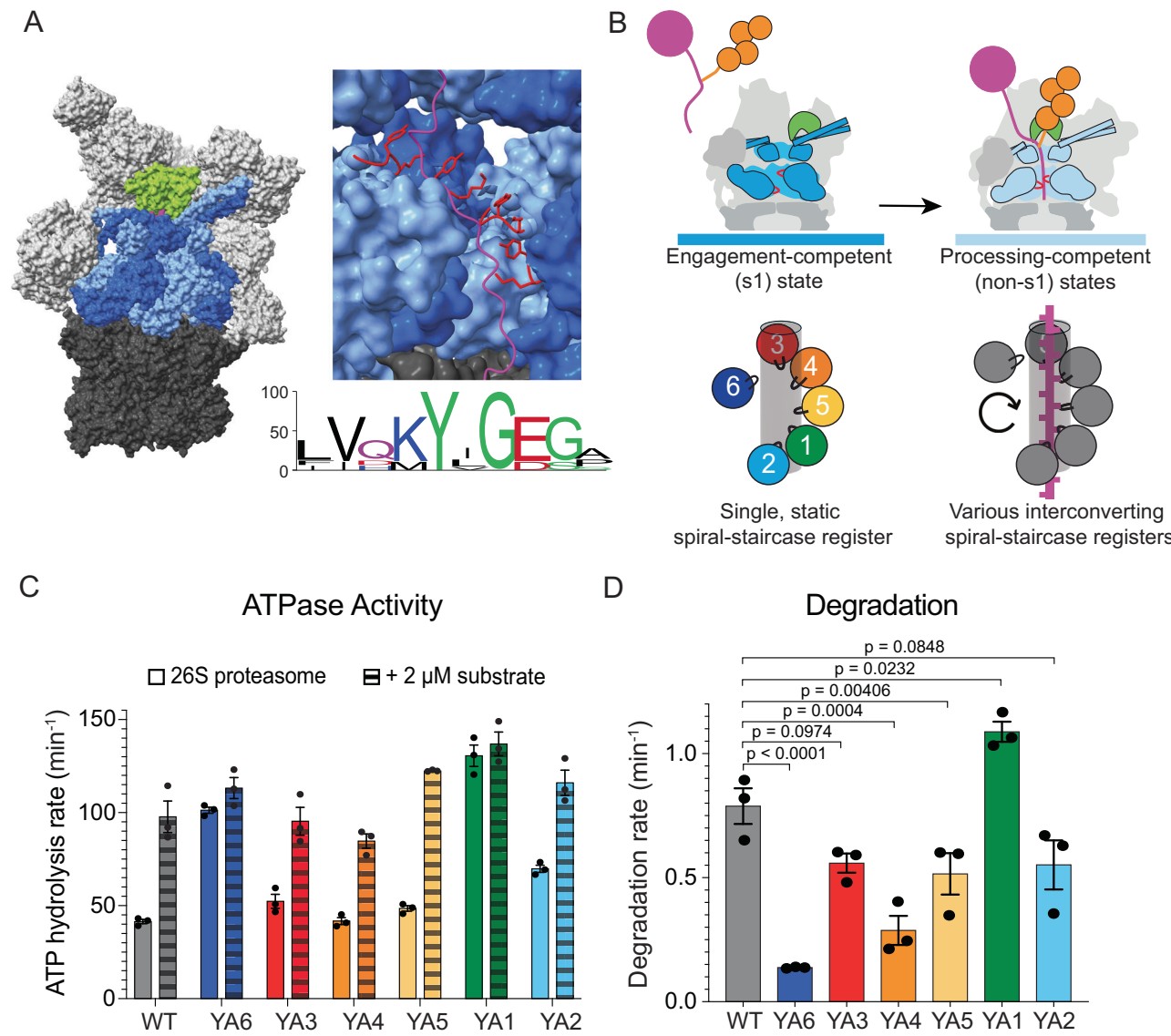

**Fig. 1 | Pore-1 loop mutations have differential effects on proteasome activities.** **A** Left: Structure of the 26S proteasome during substrate processing (PDB ID: 6EF3), with the 20S core particle in dark gray, the lid subcomplex and the non-ATPase subunits of the base subcomplex in light gray, the Rpn11 deubiquitinase subunit of the lid in lime green, the six Rpt ATPase subunits of the base alternating in two shades of blue, and the substrate polypeptide inside the central channel in magenta. Top right: Side view of the base subcomplex with two ATPase subunits, Rpt3 and Rpt4, removed for a better view of the central channel with an engaged substrate chain (magenta). Pore-1 loops (shown in red cartoon representation) with a conserved Lys-Tyr motif (or Met-Tyr for Rpt5) project from each subunit into the central channel and surround the substrate in a spiral staircase conformation. Bottom right: Sequence logo for the pore-1 loops of the yeast Rpt subunits shows the strong conservation of the Lys-Tyr motif. **B** The pore-1 loops form a specific spiral-staircase arrangement in the engagement-competent s1 state (dark blue motor), with Rpt3 at the top, Rpt2 at the bottom, and Rpt6 at the intermediate "seam" position. Upon substrate engagement, the regulatory particle shifts to processing competent non-s1 states (light blue motor) with various spiral-staircase registers that may represent intermediates of a hand-over-hand mechanism in which subunits move from the bottom via the seam position to the top of the staircase. **C** Rates for ATPase activity of reconstituted proteasomes in the absence (solid bars) and presence of 2 μM ubiquitinated titin I27[V15P] substrate (dashed bars). Technical replicates ($n = 3$) are plotted with error bars representing the SEM. **D** Rates for the degradation of FAM-labeled ubiquitinated titin I27[V15P] substrate by reconstituted proteasomes under single turnover conditions, measured by changes in fluorescence polarization ($n = 3$ technical replicates, error bars represent the SEM). Statistical significance was calculated using a one-way ANOVA test.

that in the hexamer forms a N-terminal domain ring (N-ring), and a C-terminal ATPase domain consisting of a large and a small AAA+ subdomain that assemble into the ATPase motor ring[10]. This ATPase ring docks on top of the 20S CP and induces opening of the axial gate for substrate access to the proteolytic chamber[11]. Each ATPase domain contains conserved motifs for ATP binding (Walker A) and ATP hydrolysis (Walker B), as well as a pair of pore loops, called the pore-1 and pore-2 loops, that project into the central channel of the hexameric ring[12]. The pore-1 loops sterically contact the substrate polypeptide through conserved lysine (or methionine in Rpt5) and tyrosine

residues for ATP-hydrolysis-driven substrate unfolding and translocation[11,13–16].

To be degraded by the 26S proteasome, a substrate must be modified with several ubiquitins, usually in the form of a polyubiquitin chain, and contain an unstructured initiation region for engagement by the ATPase motor[17–20]. Substrate-attached ubiquitin chains are recognized by one or more ubiquitin receptors of the proteasome, followed by insertion of the unstructured initiation region into the central channel of the ATPase hexamer. Cryo electron-microscopy (cryo-EM) studies showed that in the absence of substrate, the

proteasome primarily resides in an engagement-competent conformation, or s1 state, in which the entrance to the central channel is accessible[21–29]. Upon substrate insertion and engagement by the motor, the proteasome undergoes a conformational change from s1 to non-s1, processing-competent states. During this transition, the lid shifts and rotates relative to the base, and the Rpn11 deubiquitinase becomes coaxially aligned above the central channel, such that it partially obstructs the entrance and leaves only a small gap for the engaged substrate to be pulled through. Due to this obstruction by Rpn11, non-s1 states are unable to efficiently engage an incoming substrate, and the proteasome appears to rely on the s1 state for initial substrate insertion prior to the conformational switch. ATP-dependent translocation of the substrate by the AAA+ motor in non-s1 states then drives co-translocational deubiquitination by Rpn11[30], mechanical unfolding, and transfer of the unstructured polypeptide into the CP's internal degradation chamber for proteolytic cleavage.

Cryo-EM structures of the proteasome in the absence and presence of substrate revealed that the six Rpts and their pore-1 loops form different ordered spiral-staircase arrangements (Fig. 1B)[21–29,31–37]. The engagement-competent s1 state is characterized by a single staircase register, with Rpt3 at the top, Rpt2 at the bottom, and Rpt6 at a "seam" position, located between the top and bottom subunits. In contrast, structures of the substrate-engaged proteasome in processing-competent, non-s1 states showed various spiral-staircase registers with different subunits at the top, bottom, and seam positions, and different nucleotide occupancies[31,37]. In these non-s1 states, three to five subunits toward the top of the staircase were found engaged through their pore-1 loops with the substrate polypeptide in the central channel. These observations led to a model for sequential hand-over-hand translocation[31], in which the penultimate subunit in the staircase hydrolyzes ATP, causing the neighboring ADP-bound bottom subunit to disengage from the substrate, move as the "seam" subunit to the top of the staircase, exchange ADP for ATP, and re-engage with the substrate. The bottom or "seam" subunits in the staircase arrangements of s1- and non-s1-states are therefore expected to be the next Rpt to bind and pull on the substrate, and for the s1 state, this step may be critical for stable substrate engagement and inducing the conformational switch to non-s1 processing conformations. However, these models are solely based on structural snapshots of substrate-free and substrate-engaged proteasomes, and detailed biochemical and biophysical analyses are necessary to test these hypotheses and elucidate the principles of substrate engagement and ATP-hydrolysis-driven translocation.

Through ensemble and single-molecule Förster resonance energy transfer (smFRET) assays[17,38], we previously showed that switching from the engagement-competent s1 to the processing-competent non-s1 states upon substrate insertion into the ATPase motor is a key step for the commitment to degradation, and mutations shifting the conformational equilibrium toward non-s1 states inhibit substrate engagement and processing[34,39]. Earlier biochemical studies also attempted to determine the importance of individual pore-1 loops for substrate processing. When pore-1 loop tyrosines were individually replaced with glycine in *S. cerevisiae*, the Rpt4 mutant showed the highest accumulation of ubiquitinated conjugates in whole-cell lysates[40]. In contrast, when substituting the pore-1 loop tyrosine with an alanine (referred to as YA mutation) in yeast cells, mutant Rpt1 (YA1) and Rpt6 (YA6) exhibited the most significant growth defects at different temperatures[13]. Our own in vitro biochemical studies of YA mutants found differential defects in substrate degradation, ATP hydrolysis, and CP gate opening, with YA1 being the least affected mutant[11].

These variable defects previously observed for pore-1 loop mutants in vivo and in vitro did not reveal a clear picture of the functional contributions of individual Rpts, and a more detailed characterization of mutant effects on the various degradation steps

and the proteasome conformational dynamics is therefore needed in order to derive a conclusive mechanistic model for ATP-dependent substrate processing by the 26S proteasome.

Here, we used smFRET-based assays to conduct a comprehensive mechanistic dissection of the pore-1 loop functional asymmetries in the proteasomal AAA+ ATPase motor. Our experiments revealed that the ATPase subunits located at the bottom or "seam" positions in the Rpt spiral-staircase arrangements of substrate-free and substrate-degrading proteasomes are particularly important for substrate capture, robust unfolding without release, processive translocation, and controlling the proteasome conformational transitions. Furthermore, our cryo-EM structure determination of the substrate-free 26S proteasome revealed previously unidentified interactions between the seam subunit Rpt6 and the neighboring Rpt3 subunit, in which Rpt6's pore-1 loop is held in a helical conformation through unusual contacts with Rpt3's arginine fingers that normally coordinate the nucleotide. In this structure, we also observed that the pore-1 loop tyrosines of four ATPase subunits form stabilizing hydrogen bonds with Rpt3, explaining the static staircase and potential lack of ATP hydrolysis in the resting s1 state of the 26S proteasome. These findings point to a mechanism for substrate engagement and mechanical unfolding in which the intrinsic asymmetry of the proteasomal motor and specific spiral-staircase arrangements of Rpts play critical roles at the different stages of ATP-dependent degradation.

## Results
### Pore-1 loop mutations cause differential degradation defects
To understand how the individual pore-1 loops of Rpt1–Rpt6 contribute to substrate processing, we recombinantly expressed *Saccharomyces cerevisiae* (*S.c.*) base subcomplexes with single Tyr to Ala (YA) Rpt pore-1-loop mutations in *E. coli* (Supplementary Fig. 1A). The purified base subcomplexes were in vitro reconstituted with recombinantly expressed *S.c.* lid and endogenous 20S CP isolated from *S.c.* to form 26S proteasomes. Using recombinant systems for the lid and base subcomplexes not only gives us the advantage of introducing mutations that would otherwise be detrimental to proteasome activity in yeast, but also enables the incorporation of unnatural amino acids for site-specific labeling with fluorescent dyes to perform smFRET-based measurements of substrate processing and proteasome conformational changes[11,17,38]. As a substrate, we employed a previously characterized non-fluorescent model protein that can be labeled with fluorescent dyes for anisotropy and FRET-based measurements, and is thus well suited for degradation experiments in bulk and at the single-molecule level[17,38]. This substrate consists of a titin I27 domain with or without a destabilizing V15P mutation (titin I27$^{V15P}$ or titin I27), followed by a 24 amino acid linker that contains a single lysine for enzymatic ubiquitin-chain attachment, and a 35 amino acid unstructured initiation region or "tail" with a cysteine for labeling with a fluorescent dye (Supplementary Fig. 1B).

To test the functional defects of pore-1-loop mutant proteasomes, we first measured their ATP-hydrolysis and substrate-degradation activities. It was previously shown that the addition of substrate to wild-type proteasomes causes a 2-fold stimulation of basal ATPase activity[11], likely linked to the conformational switch from the engagement-competent s1 to processing-competent non-s1 states upon substrate engagement. Using a NADH-coupled assay to measure ATP hydrolysis, we observed the expected ~2-fold stimulation for reconstituted wild-type proteasomes, with the basal ATPase rate of $41 \pm 1$ min$^{-1}$ increasing to a rate of $98 \pm 8$ min$^{-1}$ in the presence of 2 μM ubiquitinated titin I27$^{V15P}$ (Fig. 1C, Supplementary Fig. 2A). The YA3, YA4, YA5, and YA2-mutant proteasomes showed overall similar basal ATPase rates and stimulations by substrate of 1.7–2.5 fold. In contrast, YA6 and YA1 mutant proteasomes had strongly elevated basal ATPase rates of $101 \pm 2$ and $131 \pm 6$ min$^{-1}$, respectively (Fig. 1C), that did not significantly increase upon substrate addition. An earlier study

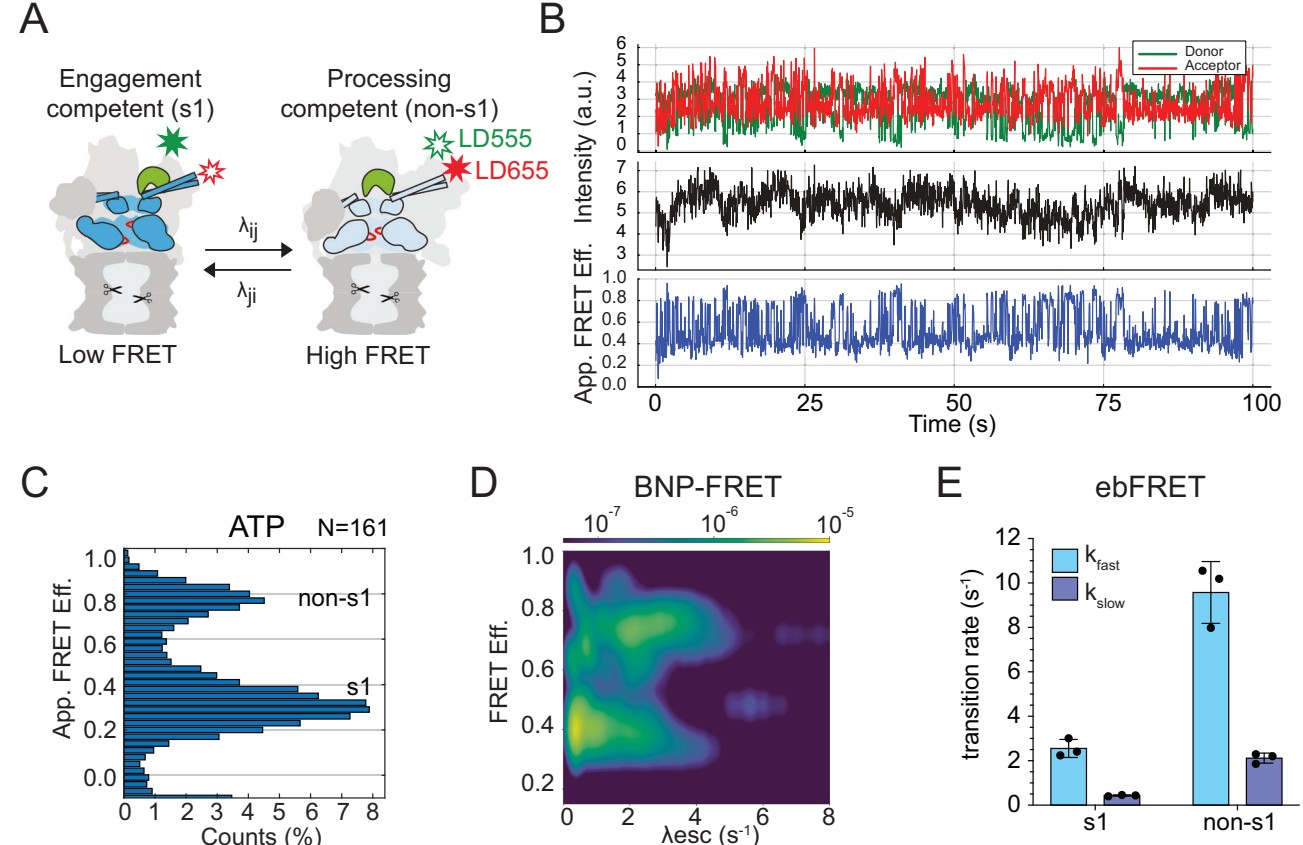

**Fig. 2 | Conformational dynamics of the 26S proteasome in the absence of substrate. A** FRET-based assay to monitor the proteasome conformational transitions between the engagement-competent s1 and processing-competent non-s1 states. Proteasomes were labeled with a FRET-donor dye (LD555, green star) on the lid subunit Rpn9 and a FRET-acceptor (LD655, red) on the base ATPase subunit Rpt5. **B** Representative traces for the single-molecule fluorescence measurement of the wild-type 26S proteasomes in ATP by TIRF microscopy. The top panel shows the fluorescence intensities for the FRET donor (green) and acceptor (red), the middle panel shows the apparent FRET efficiency, and the bottom panel illustrates the estimated most-probable state trajectory. **C** Apparent FRET efficiencies distribution for wild-type 26S proteasome in ATP generated using Spartan software ($N = 161$). **D** Bivariate probability distribution showing the escape rates ($\lambda$esc) for each FRET efficiency of wild-type 26S proteasomes in ATP ($N = 50$), as calculated using the BNP-FRET algorithm. To distinguish probability regions, a logarithmic scale color map was used (top bar), indicating the probability of proteasomes to assume a specific FRET efficiency based on all the states sampled. **E** Rates for the s1-to-non-s1 and non-s1-to-s1 transitions determined using ebFRET, performed as previously described[38], and fitting the survival plots to a double exponential ($n = 3$ biological replicates with separate ebFRET analyses, contributing to $N = 289$ particles; error bars represent the SD).

---

reported strongly increased ATPase activity for the YA1 mutant purified from yeast[13], yet this phenotype was not previously described for YA6.

To test degradation defects of the YA-mutant proteasomes, we measured their kinetics for the single turnover of the titin I27[V15P] substrate that was labeled with a N-terminal peptide modified with fluorescein amidite (FAM) for degradation readout by a decrease in fluorescence polarization (Fig. 1D). If all six pore-1 loops in the substrate-engaged, non-s1 state proteasome contributed equally to unfolding and translocation, as proposed by regular hand-over-hand translocation and sequential ATP-hydrolysis models, one would expect similar defects for all YA mutants. Although we observed that the YA3, YA5, and YA2 mutations had comparable effects (Fig. 1D, Supplementary Fig. 2B,C), YA6 and YA4 proteasomes exhibited much more decreased degradation activities of 20% and 35%, respectively. Interestingly, the YA1 mutant, which, like YA6, showed an increased basal ATPase rate with no further stimulation upon substrate addition, degraded the model substrate at 135% of the wild-type rate. YA1-mutant proteasomes can thus use their stimulated ATPase rate for faster substrate degradation, possibly because the removal of this pore-1 loop Tyr reduces crowding in the central channel and allows other subunits to move more rapidly, while not significantly reducing the motor's grip on the substrate polypeptide. In contrast, a considerable fraction of hydrolysis events for the YA6 mutant are futile. Our bulk degradation data are consistent with previous in vivo studies, indicating a particular importance of Rpt4, Rpt6, and Rpt1 for substrate processing[13,40]. However, these results do not provide mechanistic insights into how individual pore-1 loops contribute to the different substrate-processing steps and how the proteasome may utilize its structural asymmetries for efficient degradation. We therefore further investigated the functional defects of pore-1-loop mutant proteasomes, with a particular focus on Rpt4 and Rpt6.

**Pore-1 loops of Rpt4 and Rpt6 influence the proteasome conformational dynamics**

Given the strong effects of the YA4 and YA6 mutations on ATP-hydrolysis and substrate-degradation rates, we turned for further investigation to our previously developed smFRET-based assay that allows monitoring the conformational states of the 26S proteasome[38]. In this assay, the lid subunit Rpn9 is labeled with the FRET-donor fluorophore LD555 at an azido-phenylalanine (AzF) replacing Phe2, and the ATPase subunit Rpt5 is labeled with the FRET-acceptor fluorophore LD655 at an AzF substituted for Gln49 (Fig. 2A). The proteasome conformational switch from the s1 to non-s1 states leads to a >30 Å

decrease in the distance between the fluorophore-attachment points and can therefore be observed by changes in apparent FRET efficiency (app. FRET eff.)[17]. Reconstituted proteasomes were immobilized on a microscope coverslip through biotin–neutravidin interactions as previously described[38], and changes in the donor and acceptor fluorescence intensities were monitored by total internal reflection fluorescence (TIRF) microscopy (Supplementary Fig. 3). Compared to our previously published studies, we used an upgraded system with an increased signal-to-noise ratio and a faster sampling rate of 20 Hz, and we therefore first re-examined the conformational dynamics of the wild-type proteasome.

The overall FRET-state distributions of acquired traces were analyzed in apparent FRET efficiency histograms (Fig. 2C), as previously described. To calculate the conformational transition rates for proteasomes in the absence and presence of substrate, we utilized a previously described hidden Markov model technique (ebFRET) and complemented it with a recently developed Bayesian nonparametrics (BNP) FRET-analysis algorithm, BNP-FRET[41,42] (Supplementary Fig. 4). When using the ebFRET code, we pre-assigned the number of expected FRET states. ebFRET fits individual traces based on this assigned number of states and identifies the prevalent apparent FRET efficiencies. The dwell time distribution in each state can then be used to calculate kinetic transition rates (Supplementary Fig. 4A). On the other hand, the BNP-FRET algorithm implements a Bayesian non-parametric framework (Supplementary Fig. 4B) without pre-determining the number of proteasome states, and it generates probability distributions over all parameters of interest, including the number of states, transition rates, FRET efficiencies, and state trajectories obtained by drawing large number of samples using Markov chain Monte Carlo (MCMC) techniques. Furthermore, BNP-FRET accounts for photon shot noise, camera noise, and spectral crosstalk in a physically accurate manner to quantify uncertainty and complement deterministic ebFRET estimates. Specifically, with each acquired smFRET trace (Fig. 2B), the BNP-FRET algorithm explores hundreds of possible state trajectories for estimating the most probable one (Supplementary Fig. 4B, step 2), and all explored trajectories are subsequently pooled to generate ensemble-level bivariate probability distributions. These bivariate probability distributions allow a visualization of the calculated rates for the escape from each FRET efficiency ($\lambda_{esc}$ in s$^{-1}$, Fig. 2D) and of the probabilities that proteasomes assume a specific FRET efficiency based on the states sampled (Fig. 2D, color-coded bar). In addition, both ebFRET and BNP-FRET produce transition density plots that provide information about the direction of FRET-state transitions, and hence the conformational switching of proteasome molecules (Supplementary Fig. 4). To focus on the proteasome conformational transitions, fluorescence traces showing static complexes or events of photobleaching were excluded from these analyses.

For wild-type proteasomes in ATP, we observed most particles in the low-FRET, engagement-competent s1 state, with a smaller population sampling high-FRET processing-competent non-s1 states (Fig. 2C). Based on the bivariate probability distributions from BNP-FRET analyses (Fig. 2D, Supplementary Table 1), we note that kinetic rates are widely distributed, albeit with two dominant FRET efficiencies. The existence of two states is therefore not immediately evident. Indeed, we consistently observe trace-to-trace kinetic heterogeneity, indicating the possibility of a rugged energy landscape for conformational changes, making it challenging to identify states at the ensemble level. This may in part originate from the fact that there are several non-s1 states, which differ in their AAA+ motor conformations, but are degenerate in FRET efficiency. We find that wild-type proteasomes escape from the low-FRET-efficiency state (corresponding to the s1-to-non-s1 transition) with rates of 0.34–4.22 s$^{-1}$, whereas the range of rates for escape from the high-FRET-efficiency state (corresponding to the non-s1-to-s1 transition) was 0.66–7.22 s$^{-1}$. According to

the ebFRET analysis (Fig. 2E), wild-type proteasomes escape the low-FRET-efficiency state at rates of $k_{fast} = 2.55 \pm 0.41$ and $k_{slow} = 0.40 \pm 0.03$ s$^{-1}$. In contrast, the rates for escape out of the high-FRET state were $k_{fast} = 9.57 \pm 1.39$ and $k_{slow} = 2.12 \pm 0.23$ s$^{-1}$. For both transition processes, the fast rates are dominant, and we therefore largely disregarded the slow transitions whose molecular origins remain unknown. The fast rates determined by ebFRET and BNP-FRET show overall similar trends, and the observed differences are due to the trace-to-trace heterogeneity, which ebFRET cannot capture as it fits a single deterministic kinetic model meant to describe the entire dataset.

After establishing our BNP-FRET data-analysis method for the wild-type proteasome, we collected and analyzed traces for all YA mutants in the presence of either ATP (Fig. 3, Supplementary Figs. 5–8) or the non-hydrolysable ATP analog ATPγS that stabilizes the non-s1, high-FRET states (Supplementary Fig. 9). YA1, YA2, YA3, YA4, and YA5-mutant proteasomes showed FRET-efficiency distributions similar to wild type (Fig. 3A, Supplementary Fig. 6), and the degradation-deficient YA4 mutant had also conformational escape rates similar to wild-type (Fig. 3C, F–H, Supplementary Fig. 7, Supplementary Table 2), indicating that these pore loops play no major role for the intrinsic rate of the proteasome conformational switching. In contrast, the YA6-mutant proteasome exhibited an increased sampling of the processing-competent non-s1 states, as observed in the apparent FRET efficiency histogram (Fig. 3B). This was confirmed in the BNP-FRET analysis by a redistribution of the s1-to-non-s1 escape towards higher rates that extend beyond 6 s$^{-1}$ (Fig. 3E, Supplementary Table 3). Likewise, this increase was observed in the ebFRET analysis, where we determined an elevated $k_{fast}$ of $3.80 \pm 0.46$ s$^{-1}$ (compared to $k_{fast} = 2.55 \pm 0.41$ s$^{-1}$ for wild-type) for the s1-to-non-s1 escape rate, and a decrease of the non-s1-to-s1 escape rate to $k_{fast} = 6.66 \pm 0.82$ s$^{-1}$ (compared to $k_{fast} = 9.57 \pm 1.39$ s$^{-1}$ for wild type, Fig. 3G, H, Supplementary Table 4). The more frequent sampling of processing-competent non-s1 states by the YA6 mutant in the absence of substrate is also consistent with its higher basal ATPase rate. Interestingly, the BNP-FRET bivariate distribution analysis not only detected faster switching for the YA6 mutant, but also revealed a larger number of traces with decreased excursions from s1 to non-s1 states (Supplementary Tables 1 and 3, population weight 0.23 for wild type vs. 0.31 for YA6 proteasomes), suggesting that Tyr removal from Rpt6's pore-1 loop can both accelerate and decelerate the conformational dynamics of the proteasome. In contrast, the YA1-mutant proteasome did not show such a shift in the conformational distribution (Supplementary Fig. 6), and its increased basal ATP hydrolysis may be caused by an effect distinct from that observed for the YA6 mutant. It is possible that removal of Rpt1's Tyr allows for a less static s1 state that hydrolyzes ATP at a considerable rate without globally switching to non-s1 conformations. Based on our data, Rpt6 appears particularly important for controlling the proteasome's conformational transitions. Since substrate engagement triggers the switch from s1 to non-s1 states and Rpt6 resides at the seam position in the s1-state spiral staircase, this subunit and its pore loop may be important for sensing the absence or presence of a substrate polypeptide in the central channel. Structures of various other AAA+ protein translocases in the presence of substrate revealed a network of interactions between conserved pore-1 loop residues, in which a positively charged Lys or Arg preceding the grip-conferring tyrosine interacts with a negatively charged Glu or Asp in the pore-1 loop of the neighboring ATPase subunit in the ring[43]. Furthermore, pore-1-loop interactions were suggested to gate the central channel prior to substrate insertion, and it is possible that mutation of Rpt6's tyrosine disrupts these interactions and partially decouples s1-to-non-s1 conformational changes from substrate engagement.

To assess this further, we solved the high-resolution cryo-EM structure of the substrate-free s1-state proteasome (Fig. 4, Supplementary Figs. 10, 11). Surprisingly, we observed that Rpt6's pore-1 loop

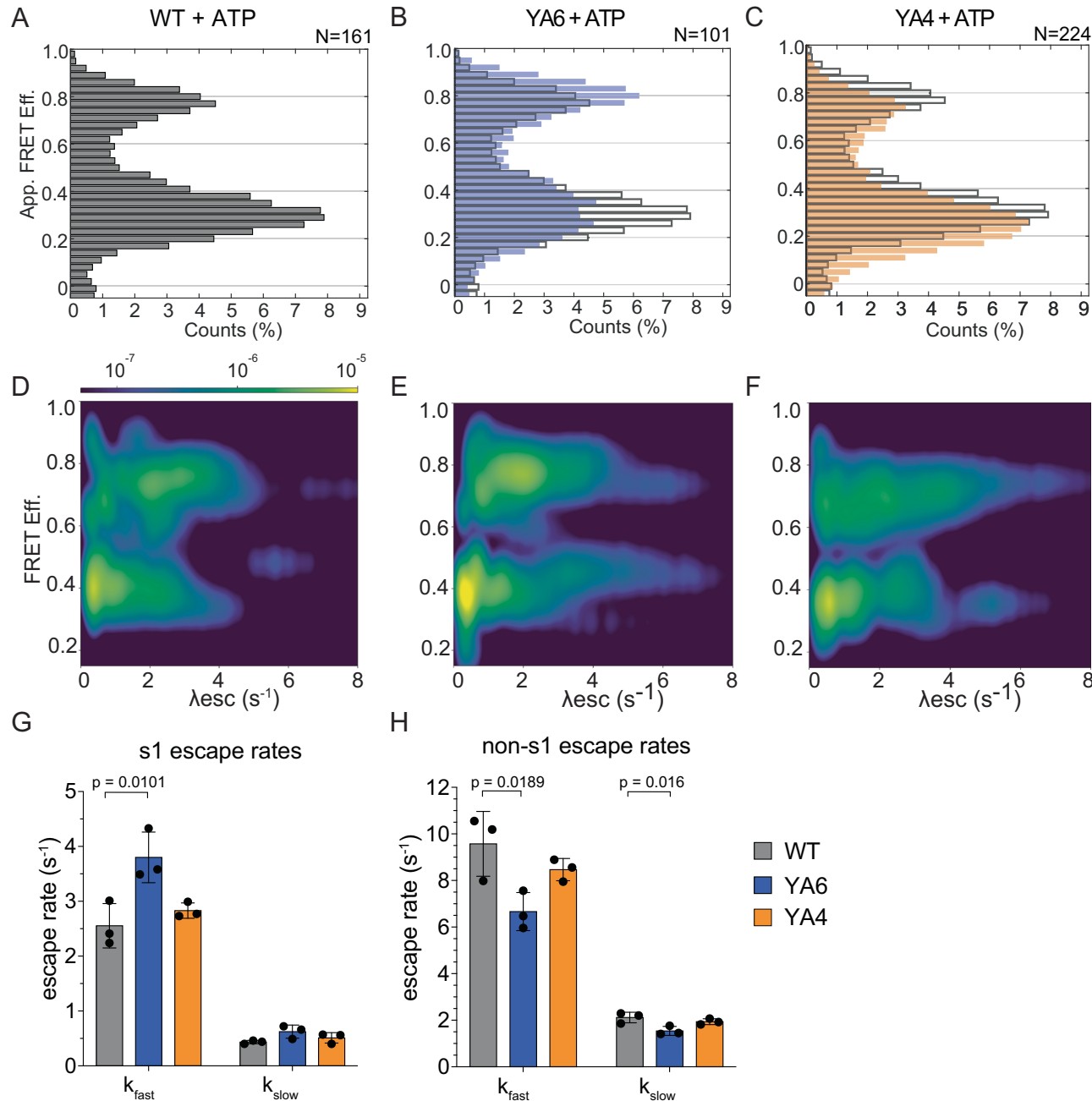

**Fig. 3 | Differential effects of Rpt4 and Rpt6 pore-1 loop mutations on the proteasome conformational dynamics. A** Apparent FRET-efficiency distribution for wild-type proteasomes in ATP. **B, C** Apparent FRET efficiency distributions for YA6 (blue bars) and YA4 (orange bars) mutant proteasomes overlayed with the distribution of wild-type proteasome (gray outlined bars). **D–F** BNP-FRET derived bivariate probability distributions for the escape rates ($\lambda_{esc}$) of wild-type, YA6, and YA4 mutant 26S proteasomes in ATP, respectively. The logarithmic-scale color map (see top bar) describes probability regions that indicate the likelihood of proteasomes to assume a specific FRET efficiency based on all sampled states. **G, H** Escape rates determined using ebFRET, performed as previously described[38] and fit to a double exponential ($n = 3$ technical replicates contributing to a total of 289 particles for WT, 175 for YA6, and 361 for YA4. Error bars represent the SD between technical replicates.) Statistical significance was calculated using a one-way ANOVA test.

is tucked away from the central channel and adopts a helical conformation that is stabilized through interactions with the arginine-finger residues of the neighboring Rpt3 subunit (Fig. 4C, D, Supplementary Fig. 11). Normally, the arginine fingers of a particular ATPase subunit coordinate the nucleotide bound to the counterclockwise-next subunit and facilitate hydrolysis. The interactions observed for the Rpt6/Rpt3 interface may be responsible for "parking" Rpt6's pore-1 loop and stabilizing the specific s1-state spiral-staircase conformation, with Rpt3 at the top and Rpt6 in the seam position, until a substrate is

inserted into the central channel. In fact, our structure suggests that the pore-1 loop tyrosines of four ATPase subunits, Rpt1, Rpt2, Rpt6, and Rpt3, form hydrogen bonds with backbone carbonyls or side chains of Rpt3, which may hold the s1 state static (Fig. 4B). While the tyrosines of Rpt1 and Rpt2 contact Asn285 and the backbone of Phe282 in Rpt3's pore-2 loop, respectively, Rpt6's tyrosine interacts with Asp325 in Rpt3's second region of homology (SRH), and Rpt3's tyrosine itself is held in place by contacting the backbone carbonyl of Thr114 in Rpt3's N-terminal domain as part of the rigid N-ring. These

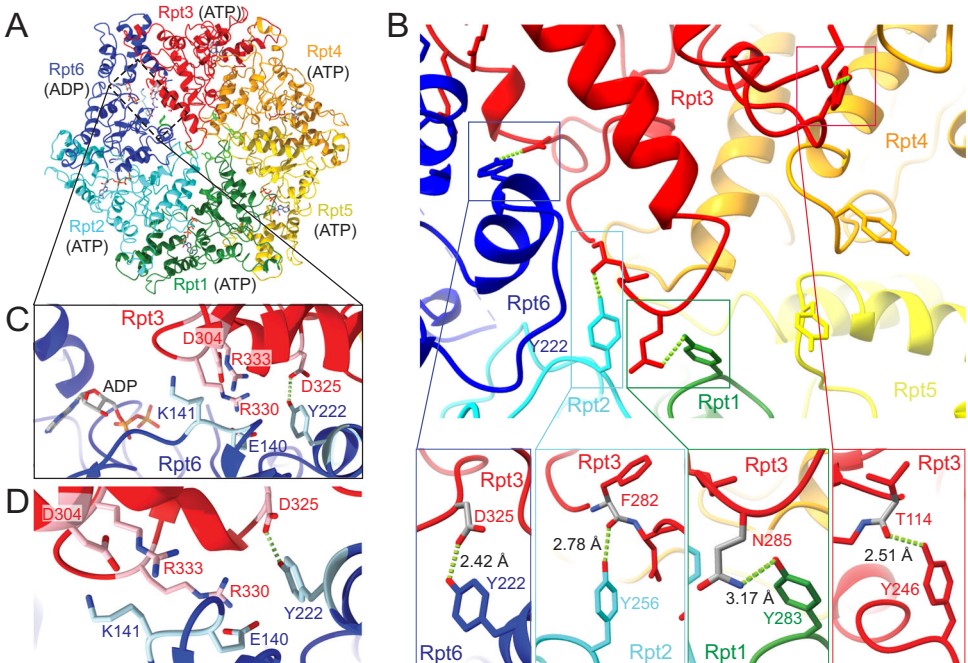

**Fig. 4 | Distinct interactions stabilize Rpt6's pore-1 loop and three other pore-loop Tyr in the ATPase hexamer of the s1 state. A** Top view of the atomic model for the Rpt hexamer derived from the cryo-EM structure of the yeast 26S proteasome in the processing-competent s1 state. The Rpt subunits are in a spiral-staircase arrangement with Rpt3 (red) at the top, Rpt2 (cyan) at the bottom, and Rpt6 (blue) in the seam position. All subunits are bound to ATP, except for the ADP-bound Rpt6. Nucleotides, the pore-1 loop tyrosines (green), and Rpt3's arginine fingers at the Rpt6/Rpt3 interface are shown in stick representation. **B** Top: Zoom-in on the central channel shows that the pore-1 loop tyrosines of Rpt1, Rpt2, Rpt6, and Rpt3 appear to form hydrogen bonds with the backbone or side chains of Rpt3, likely stabilizing this spiral-staircase register and preventing ATP hydrolysis or conformational changes in the s1 state. Bottom: Detailed views of individual interactions between pore-1 loop tyrosines and residues of Rpt3, with potential hydrogen bonds indicated by green dashed lines and measurements given for the C–C or C–N distances, which lie in an ideal range for hydrogen bonding. **C** Zoom-in on the Rpt6/Rpt3 subunit interface, showing how Rpt6's pore-1 loop around Y222 adopts a helical conformation that appears to be stabilized through interactions with Rpt3's D325, R330, and R333. E140 of Rpt6 seems to mediate the interactions with the R330 and R333 arginine fingers that are far displaced from the Rpt6-bound nucleotide they normally coordinate during the ATP-hydrolysis cycle. **D** Alternative perspective on the Rpt6/Rpt3 interface to highlight the interactions of Rpt6's E140 and K141 with Rpt3's R330, R333, and D304.

interactions would explain why only a single spiral staircase arrangement is present in the s1 state and why ATP hydrolysis is significantly reduced, potentially even completely shut down, while the proteasome resides in this state. The basal ATPase activity measured for substrate-free yeast proteasomes may thus originate primarily from spontaneous transitions to non-s1 states, in which ATP hydrolysis leads to the progression of the Rpt ring through different staircase registers. These transitions may represent stochastic thermal crossings of the energy barrier between s1 and non-s1 states in this conformational equilibrium of the yeast proteasome. Interestingly, the pore-1 loop interactions stabilizing the resting, engagement-competent s1 state are also present in the human 26S proteasome (Supplementary Fig. 12A[44]), where the pore-1 loop tyrosines of Rpt1, Rpt2, and Rpt3 show equivalent hydrogen-bond interactions. Although the aromatic residues in the pore-1 loop of human Rpt6 are a phenylalanine and therefore unable to form a hydrogen bond with Rpt3, the pore-1 loop itself shows a similar helical conformation as in the yeast proteasome and appears to be stabilized through electrostatic interactions with Rpt3's SRH and pore-2 loop. The human proteasome in the absence of substrate has undetectable ATPase activity, and based on cryo-EM particle distributions, it rarely switches spontaneously to processing states[44], supporting our hypothesis that the resting, s1 conformation of the 26S proteasome is inactive in ATP hydrolysis and held static, primarily due to the special interface between Rpt6 and Rpt3. The YA mutation in Rpt1 of the yeast proteasome may disrupt this static conformation and allow some ATP hydrolysis in the s1 state without inducing a global shift of the conformational equilibrium toward non-s1 states (Fig. 1C, Supplementary Fig. 6). In contrast, the YA mutation in Rpt6 likely disrupts the interactions that hold the Rpt6 pore-1 loop in a parked

position and hence leads to more frequent spontaneous switching to non-s1 states with correspondingly higher ATPase rates. Importantly, none of the other Rpt interfaces in the s1-state proteasome show interactions similar to those between Rpt6's pore-1 loop and Rpt3's Arg fingers (Fig. 4C, D), and such contacts are also not observed in substrate-processing, non-s1 states, neither at seam- nor non-seam interfaces (Supplementary Fig. 12B)[31]. Rpt6's pore-1 loop interactions with the arginine fingers R330 and R333 of Rpt3 appear to be mediated by Rpt6's E140, which is flanked by K141 that may contribute to correctly positioning these charged residues through interactions with Rpt3's D304 (Fig. 4D). To confirm those interaction and the role of the Rpt6/Rpt3 interface in stabilizing the s1 state, we created proteasome variants with Rpt6 E140A or K141A mutation and measured their ATP-hydrolysis and substrate-degradation activities. The E140A and K141A mutants showed increased basal ATPase rates of $108 \pm 7$ and $87 \pm 14$ min$^{-1}$, respectively, about two-fold higher than for the wild-type proteasome (Supplementary Fig. 13A), which is consistent with a disruption of the s1 state and a conformational shift to the hydrolysis-active non-s1 states. This was confirmed by single-molecule FRET analyses, in which the Rpt6 E140A mutant proteasome showed a marked shift in the conformational distribution from s1 to non-s1 states (Supplementary Fig. 13B). Yet, neither the E140A nor the K141A mutant showed significant changes in degradation velocity (Supplementary Fig. 13C), indicating that the rate-limiting step of substrate turnover remained unaffected. These mutants were also robustly stimulated in their ATP hydrolysis by substrate engagement (Supplementary Fig. 13A), which is similar to wild type and in strict contrast to the YA6-mutant, whose lack of the pore-1 loop Tyr leads to major degradation defects.

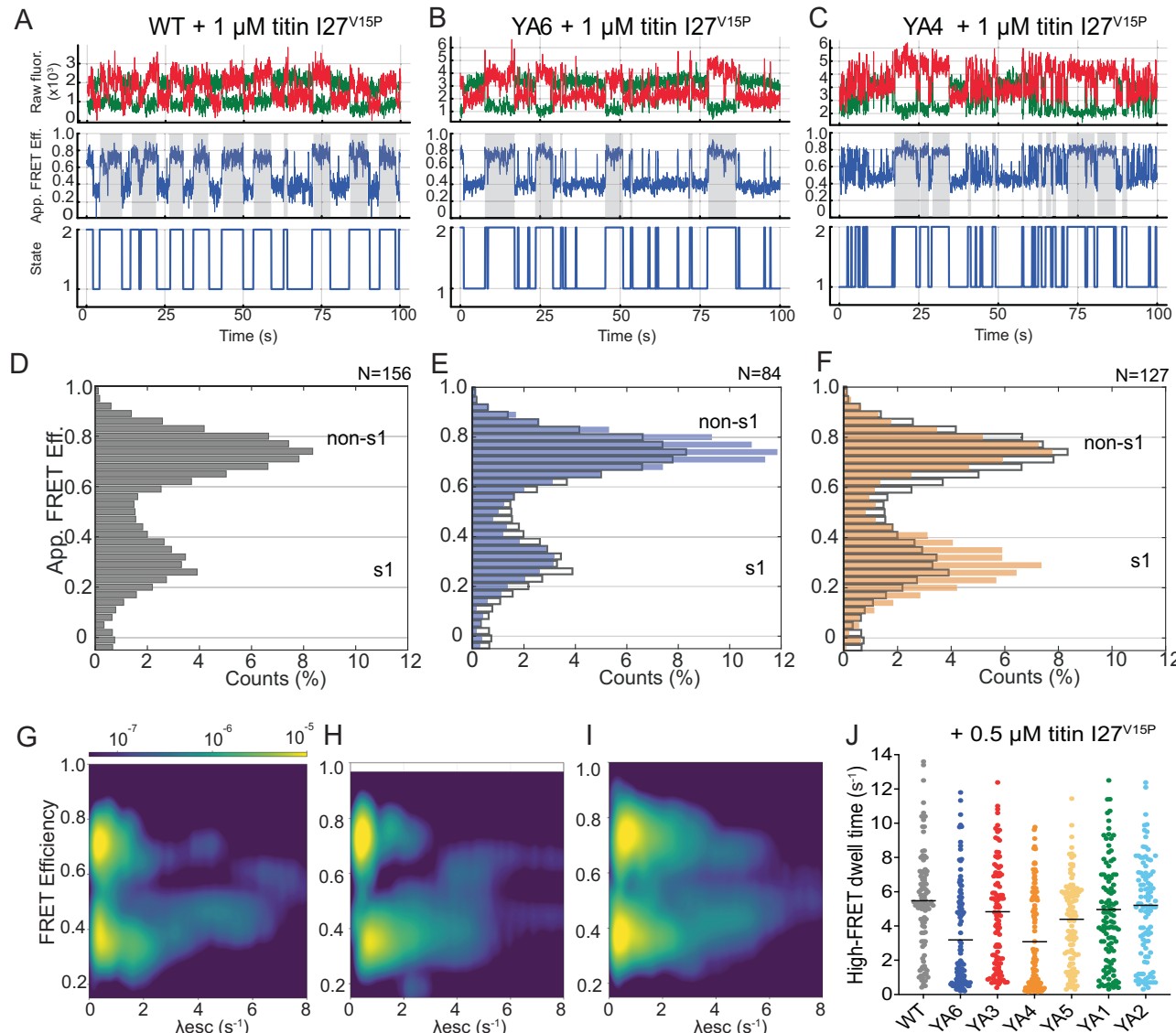

**Fig. 5 | The YA4 and YA6 mutations lead to frequent transitions out of the processing-competent states during substrate degradation.** Example traces for wild-type (**A**), YA6 (**B**), and YA4 mutant 26S proteasomes (**C**) in the presence of 1 µM ubiquitinated titin I27$^{V15P}$ substrate, with the donor (green) and acceptor (red) fluorescence intensities shown in the top panels, the apparent FRET efficiencies in the middle panels, and estimated most-probable state trajectories in the bottom panels. High-FRET dwells during substrate processing are highlighted by gray shading. **D** Apparent FRET efficiency distributions for wild-type proteasomes 1 µM ubiquitinated titin I27$^{V15P}$ substrate. **E, F** Apparent FRET efficiency distribution for YA6 proteasomes (blue bars) and YA4 proteasomes (orange bars) in the presence of 1 µM ubiquitinated titin I27$^{V15P}$ substrate, compared to the distribution for wild-type proteasomes (gray outlined bars). **G–I** Bivariate probability distributions for the escape rates ($\lambda_{esc}$) of wild-type, YA6, and YA4 proteasomes, respectively, in the presence of 1 µM ubiquitinated titin I27$^{V15P}$ substrate. The logarithmic-scale color map (see top bar) describes probability regions that indicate the likelihood of proteasomes to assume a specific FRET efficiency based on all sampled states. **J** Lengths of continuous high-FRET phases during the degradation of ubiquitinated titin I27$^{V15P}$ substrate by wild-type and YA-mutant proteasomes. The substrate concentration for these analyses was 0.5 µM for larger spacing and thus better discrimination of individual substrate-processing events. Black lines represent the median value of the distribution. For the cumulative frequency analysis of these dwell times (see Supplementary Fig. 17). Statistical analyses and number of analyzed single-molecule events from three technical replicates, each with 3–5 movies are summarized in Supplementary Table 8.

To better understand the effect of pore-1 loop mutations on substrate processing, we measured the single-molecule conformational dynamics of the proteasome in the presence of a model substrate, ubiquitinated titin I27$^{V15P}$. We previously found that upon substrate engagement, the proteasome stably switches to processing-competent, high-FRET non-s1 states that are maintained until substrate unfolding and threading through the ATPase motor has been completed[38]. We first confirmed these findings by observing high-FRET phases, i.e., substrate-processing dwells that include unfolding and complete threading, and apparent FRET efficiency distributions for the wild-type proteasome in the presence of ubiquitinated titin I27$^{V15P}$

substrate, which were comparable to the previously reported values (Fig. 5A, D; Supplementary Figs. 14–16). Traces for the YA4 mutant exhibited much shorter dwells, which were caused by transient low-FRET-state excursions interrupting the high-FRET degradation phase (Fig. 5C, F; Supplementary Fig. 14C). Our previous investigations of conformational transitions revealed that the wild-type proteasome also shows brief low-FRET excursions, with a frequency that positively correlates with the substrate's thermodynamic stability[38]. These transitions may thus represent slippage events and temporary returns to the s1 state after unsuccessful substrate-unfolding attempts by the motor.

Because the yeast 26S proteasome in the absence of substrate also has a tendency to spontaneously transition between s1 and non-s1 states, we were previously unable to use ebFRET for reliably measuring the conformational transition rates in the presence of substrate. However, the BNP-FRET method allowed us to determine those kinetics. In the presence of 1 μM titin I27$^{V15P}$ substrate, the YA6 mutant resembled the wild-type proteasome in its overall FRET-efficiency distribution (Fig. 5D, E) and in the rate ranges for the transition from the low-FRET to high-FRET states, which were 0.66–6.12 s$^{-1}$ for wild type and 0.78–7.19 s$^{-1}$ for YA6 (Fig. 5G, H, Supplementary Tables 5, 7). Interestingly, for the high-FRET to low-FRET transition, YA6 showed slightly higher rates of 0.38–8.72 s$^{-1}$, compared to 0.55–6.61 s$^{-1}$ for WT (Fig. 5G, H, Supplementary Tables 5, 7). The YA1, YA2, YA3, and YA5-mutant proteasomes showed wild-type-like FRET-efficiency distributions (Supplementary Fig. 15). In contrast, the YA4-mutant exhibited a pronounced shift in its FRET-efficiency distribution toward the low-FRET s1 state (Fig. 5F) and correspondingly higher rates for the escape from the high-FRET states (0.64–10.90 s$^{-1}$, Fig. 5I), which is consistent with the frequent excursions to the low-FRET state that we observed in individual traces during substrate processing (Fig. 5C).

The FRET-efficiency traces for proteasomes in the presence of substrate contain phases of active degradation as well as phases between degradation events, during which the proteasome is not occupied with substrate. To more specifically analyze how the YA mutations affect the conformational transitions during active degradation, we manually scored the high-FRET phases of proteasomes in the presence of 0.5 μM ubiquitinated titin I27$^{V15P}$ substrate. This lower substrate concentration reduced the frequency of degradation events and made it easier to distinguish between phases of degradation and idling without a bound substrate. The wild-type proteasome showed an average length of $\tau^{WT} = 5.5 \pm 0.3$ s for the persistence of high-FRET, processing-competent non-s1 states, and the YA1, YA2, and YA5 mutants showed similar values, whereas YA4 and YA6-mutant proteasomes had shorter high-FRET phases, with $\tau^{YA4} = 3.2 \pm 0.3$ s and $\tau^{YA6} = 3.1 \pm 0.3$ s, respectively (mean ± SEM; Fig. 5J, Supplementary Table 8). Analysis of the cumulative frequencies of dwell times in the high-FRET phase for the degrading WT and YA-mutant proteasomes revealed that the YA4 and YA6 mutants clearly deviate, with the majority of their continuous high-FRET phases lasting <2 s (Supplementary Fig. 17). We therefore hypothesize that the absence of the pore-1 loop tyrosine in Rpt4 and Rpt6 reduces the motor grip, causes slippage during substrate-unfolding attempts, and leads to recurring transitions to the s1 state, potentially accompanied by a local disengagement from the substrate in the central channel.

In summary, our data revealed that the pore-1 loop mutation in Rpt6 influences the proteasome conformational dynamics in the absence of substrate, and mutation of either Rpt6's or Rpt4's pore-1 loop compromises the stable maintenance of the processing-competent, non-s1 states during substrate unfolding and causes more frequent returns to the s1 state, potentially due to slippage.

## Pore-1 loops of Rpt4 and Rpt6 play distinct roles in substrate processing

While YA mutations in Rpt1, Rpt2, Rpt3, and Rpt5 only minimally affected substrate degradation, YA mutations in Rpt4 and Rpt6 led to major defects (Fig. 1D). We therefore focused on these two subunits and employed a previously developed FRET-based substrate-processing assay to elucidate how their pore-1 loops contribute to the individual steps of degradation[17,38]. This FRET-based assay relies on a fluorophore attached to the substrate's flexible initiation region and a second fluorophore attached to an AzF substituted for Ile191 in the linker between the N-domain and ATPase domain of Rpt1, allowing to directly monitor the progression of a substrate polypeptide through the central channel (Fig. 6A). The dyes were placed such that initial substrate binding to the proteasome leads to an intermediate FRET

efficiency, which increases to a maximum value after substrate-tail insertion, engagement by the ATPase motor, and the onset of translocation. The high FRET efficiency then persists during deubiquitination, before declining as the substrate is further translocated through the ATPase channel and into the 20S CP[38].

First, we used this assay in stopped-flow mixing experiments with sulfo-Cy3 donor-dye labeled proteasomes and sulfo-Cy5 acceptor-dye labeled ubiquitinated titin I27$^{V15P}$ substrate to monitor the kinetics of substrate-tail insertion and engagement in bulk[17]. For these ensemble measurements, reconstituted proteasomes were incubated with the Rpn11 inhibitor 1,10-phenanthroline (oPA) to stall substrate processing at the high-FRET deubiquitination phase right after engagement, leading to a single-turnover scenario in which the combined first-order time constant for substrate binding, tail insertion, and engagement could be determined by fitting the increase in FRET-efficiency (Fig. 6B, Supplementary Fig. 18). This time constant was $\tau_{ins}^{WT} = 6.8 \pm 0.1$ s for wild-type proteasomes, whereas YA4- and especially YA6-mutant proteasomes showed significant delays, with time constants of $\tau_{ins}^{YA4} = 14.3 \pm 0.8$ s and $\tau_{ins}^{YA6} = 18.0 \pm 2.5$ s, respectively. The almost 3-fold slower substrate insertion into the YA6 mutant can be explained to some extent by the shift of the conformational equilibrium and frequent spontaneous sampling of the processing-competent non-1 states, in which the Rpn11 deubiquitinase obstructs the entrance to the motor channel and thus inhibits substrate insertion. Importantly, the YA mutation likely dislodges Rpt6's pore-1 loop from its tucked-away interactions with Rpt3's Arg fingers, while, at the same time, hampering the grip on the substrate for stable engagement, such that Rpt6's mutant pore-1 loop may sterically interfere with the diffusive insertion of the substrate tail into the central channel. Interestingly, the YA6 mutant not only exhibited slower kinetics but also a 4.5-fold lower amplitude than the wild-type proteasome in the FRET-signal change, suggesting that substrates either do not fully enter the ATPase ring or fail to stay stably inserted. Lack of the Rpt6 pore-1 loop tyrosine thus not only delays substrate insertion, but also appears to compromise substrate capture, likely due to defects in motor engagement of the substrate's initiation region.

To gain more detailed insights into these engagement problems and discern potential defects in mechanical unfolding, we switched to the thermodynamically more stable wild-type titin I27 substrate. After confirming robust degradation of this substrate by wild-type and YA-mutant proteasomes in bulk (Supplementary Fig. 19), we used the FRET-based substrate-processing assay in our upgraded single-molecule TIRF microscopy setup for further investigation. In this single-molecule assay, the substrate was LD555-donor labeled, and immobilized proteasomes carried a LD655 acceptor. Fluorescence traces for this experimental setup show an initial intermediate FRET-efficiency value of ~0.5 upon binding of the ubiquitinated substrate to a proteasomal receptor, followed by a FRET increase to its maximum value of 0.8–1.0 when the substrate tail inserts into the ATPase motor, and a short high-FRET dwell during Rpn11-mediated deubiquitination (Fig. 6C). Further substrate translocation then leads to a decrease in the apparent FRET-efficiency signal to a persistent value of 0.35–0.45 that we identified as a pre-unfolding dwell based on its distinct presence for wild-type titin. This pre-unfolding dwell was much harder to observe previously when using the more labile and readily unfolded titin I27$^{V15P}$ substrate and our old microscopy setup with lower signal-to-noise. Successful unfolding and continuous translocation subsequently result in a FRET-efficiency decay to ~0.1–0.2, before proteolytic cleavage and release of the donor-labeled peptide fragment from the internal proteasome chamber leads to loss of the donor signal.

We analyzed the FRET-efficiency traces for wild-type and YA-mutant proteasomes during titin I27 degradation by scoring the processing steps represented by the individual FRET phases and plotting their cumulative distribution functions (1-CDF or survival plots, Fig. 6F–I, Supplementary Fig. 20). Fitting the survival plots for the tail

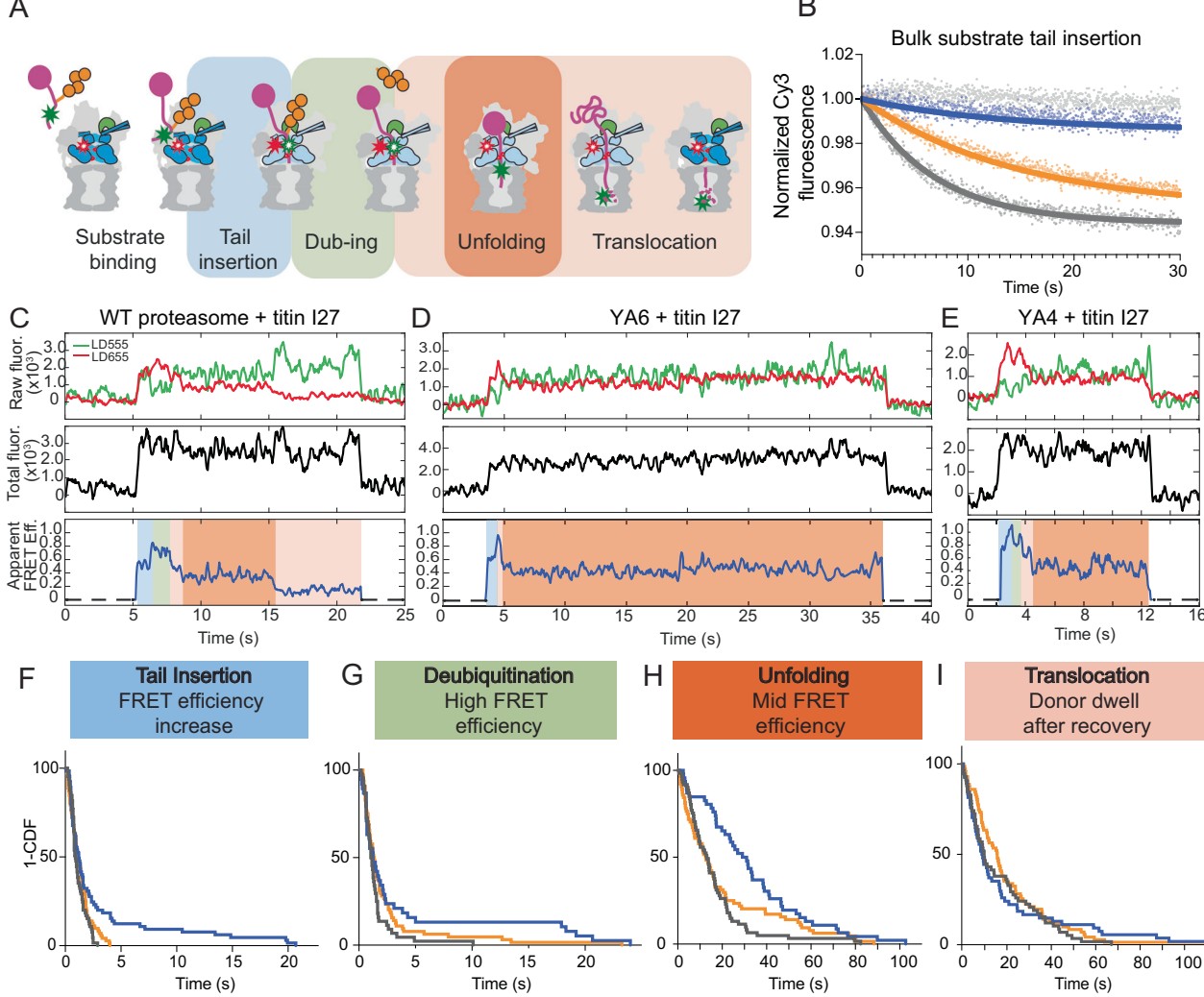

**Fig. 6 | Direct observation of substrate processing by pore-1 loop mutant proteasomes using single-molecule FRET. A** Schematic for the FRET-based substrate-processing assay, monitoring the individual steps of substrate binding and tail insertion (blue shading), deubiquitination (green shading), unfolding (dark orange shading), and translocation (light orange shading). **B** Representative traces and single exponential fits for the bulk substrate-processing assay with Cy5-acceptor-labeled substrate and Cy3-donor-labeled proteasomes whose Rpn11 deubiquitinase was inhibited to stall substrate processing after tail insertion and engagement. Shown are the changes in Cy3 donor fluorescence after stopped-flow mixing with ubiquitinated titin I27$^{V15P}$ substrate for wild-type (dark gray), YA6 (blue), and YA4 proteasomes (orange). The data were normalized to initial fluorescence values. A control with ubiquitinated, Cy5-labeled titin I27$^{V15P}$ substrate and wild-type Cy3-labeled base subcomplex alone, in the absence of lid and core peptidase, is depicted in light gray. **C** Representative single-molecule fluorescence traces for the degradation of LD555 donor-labeled ubiquitinated titin I27 substrate by LD655 acceptor-labeled 26S proteasomes. The top panel shows fluorescence intensities for the FRET donor (green) and acceptor (red), the middle panel shows

the calculated total fluorescence intensity, and the bottom panel illustrates the calculated apparent FRET efficiency. Substrate degradation events showed a constant total fluorescence, indicating the presence of a single substrate per proteasome. Traces were filtered for visualization using Matlab R2019b, moving average filter with a heuristically determined averaging window of five frames (equivalent to 250 ms). Example traces for titin I27 substrate processing by YA6 (**D**) and YA4 mutant proteasomes (**E**) show longer dwells for the unfolding phase, followed by an abrupt loss of the FRET signal without a post-unfolding translocation phase. **F** Cumulative distribution function (1-CDF) or survival plot analysis for the tail insertion and engagement step of wild-type proteasome (gray, $N = 65$ events), the YA6 mutant (blue, $N = 66$), and the YA4 mutant (orange, $N = 72$). **G** Survival plot analysis for the deubiquitination step of wild-type proteasome (gray, $N = 44$), the YA6 mutant (blue, $N = 39$), and the YA4 mutant (orange, $N = 64$). **H** Survival plot analysis for the unfolding step of wild-type proteasome (gray, $N = 61$), the YA6 mutant (blue, $N = 48$), and the YA4 mutant (orange, $N = 64$). **I** Survival plot analysis for the translocation step after the unfolding dwell for wild-type proteasome (gray, $N = 58$), the YA6 mutant (blue, $N = 54$), and the YA4 mutant (orange, $N = 71$).

insertion phase to single-exponential functions revealed that the majority of particles for all YA mutants showed wild-type-like kinetics, with time constants ranging from 0.8 to 1.6 s (Supplementary Table 9). Only a small fraction of ~10% for the YA6-mutant proteasome took on the substrate with delayed kinetics (Fig. 6F). Successful tail insertion thus occurs largely with similar kinetics even for the engagement-defective YA6 mutant, suggesting that the defects we observed in bulk measurements originate from failed insertion attempts that are not considered in the survival plots (see below). The time constants for the subsequent deubiquitination phase were again similar for all YA

mutants and wild type, ranging between 0.8 and 1.5 s (Supplementary Table 10). Interestingly, substrate unfolding also occurred at wild-type-like rates for all YA mutants except YA6, which spent 3-fold more time on unraveling titin I27 ($\tau_{unf}^{YA6} = 50.8$ s versus $\tau_{unf}^{WT} = 16.3$ s; Fig. 6D, H Supplementary Table 11). This observation for the YA6 proteasome is remarkable, as we detected no obvious unfolding defect for this mutant during degradation of the less stable titin I27$^{V15P}$ in our single-molecule conformational dynamics and substrate-processing assays (Fig. 5B, E, H; Supplementary Fig. 21, Supplementary Table 13). Manifestation of YA6's unfolding defects thus appears to depend on the

substrate's thermodynamic stability and possibly the motor's behavior during unfolding attempts. Also surprising is that the YA4 mutant showed no extended unfolding phase, suggesting that its degradation defects are not necessarily caused by slower unfolding, but possibly lower processivity and premature release (see below). The subsequent substrate-translocation phase, identified as the donor dwell after successful substrate unfolding and FRET-efficiency decay to ~0.1–0.2, was similar for wild-type and most of the pore-1 loop-mutant proteasomes, with a slightly longer time constant for YA4 and faster translocation for the YA1 mutant (Fig. 6I, Supplementary Fig. 20, Supplementary Table 12).

### Efficient substrate capture depends on both Rpt6 and Rpt4

Since the YA6-mutant proteasome exhibited substrate-tail insertion and engagement defects in our ensemble measurements that were not as obvious when monitoring successful processing events by single-molecule FRET, we analyzed the success rate of this mutant in capturing the titin I27 substrate compared to wild-type and other YA-mutant proteasomes. The capture success for wild-type and mutant proteasomes was quantified based on at least 210 events of substrate interactions, for which we calculated the ratio of successful substrate engagement and processing relative to the total number of substrate-binding events, as previously described[38]. These calculations yielded a capture-success rate of $54.7 \pm 7.6\%$ for the wild-type proteasome and similar values for the YA1, YA2, YA3, and YA5 mutants, indicating no defects in substrate engagement for these variants (Fig. 7A, Supplementary Fig. 23A, Supplementary Table 14). However, the YA6-mutant proteasome showed a capture success of only $30.2 \pm 8.3\%$, and substrate capture of the YA4 mutant was with $33.9 \pm 4.5\%$ also considerably diminished (Fig. 7A). The almost 2-fold reduced ability of the YA6 mutant to successfully engage a substrate before it dissociates from a ubiquitin receptor is likely caused by a lack of grip due to the missing pore-1-loop Tyr in Rpt6 and the observed shift in this mutant's conformational equilibrium, leading to a ~2-fold reduced abundance of the engagement-competent s1 state (Fig. 3E). In contrast, the YA4 mutant did not show an equivalent shift in the conformational equilibrium. Insights into potential differences between the underlying reasons for the substrate-capture problems of YA6 and YA4-mutant proteasomes were provided by analyzing the residence time of substrates on the proteasome during unsuccessful capture attempts. These interactions were short for the YA6-mutant with an average time constant of $\tau = 0.4$ s, whereas substrates spent ~2-fold more time on the YA4-mutant proteasome before dissociating ($\tau = 0.8$ s; Supplementary Fig. 23C). While the lack of an intact Rpt6 pore-1 loop seems to cause early defects by conformationally interfering with substrate-tail insertion or not properly guiding the substrate into the central channel, mutation of Rpt4's pore-1 loop may prevent a stable commitment to substrate degradation at a slightly later stage, for instance due to failed power strokes that drive substrate translocation after engagement. As expected, measurements with the less stable titin I27$^{V15P}$ substrate revealed a similar picture, with a capture success of $50.6 \pm 3.4\%$ for wild-type proteasome and $34.3 \pm 1.0\%$ and $37.5 \pm 2.3\%$ for the YA4 and YA6 mutants, respectively (Supplementary Fig. 23B, Supplementary Table 14).

### Rpt4 and Rpt6 ensure robust substrate unfolding without release

Interestingly, many FRET traces for titin I27 degradation by the YA4- and YA6-mutant proteasomes did not show a final translocation phase, but instead ended abruptly with a loss of the donor signal while the proteasomes were in the pre-unfolding phase (Fig. 6D, E), suggesting that unsuccessful unfolding attempts led to slippage and terminal substrate release. The hypothesis that motor slippage could lead to substrate backsliding in the central channel is supported by interesting FRET-efficiency patterns that we observed for several YA-mutants and

even the wild-type proteasome during titin I27 degradation, where multiple transient high-FRET peaks interrupted the persistent 0.35–0.45 FRET efficiency phase in the pre-unfolding dwells (Supplementary Fig. 22). This appear to be a direct observation of substrate slippage, where loss of grip by the pore-1 loops may cause the unstructured substrate tail to backslide, bringing the attached donor dye again into closer proximity to the acceptor dye above the ATPase ring.

To further investigate the contributions of individual pore-1 loops to unfolding and motor grip on the substrate, we analyzed >100 traces each for wild-type and YA-mutant proteasomes, and determined their frequency of terminal substrate release after unsuccessful unfolding attempts, i.e. events where substrate processing ended abruptly with a loss of donor fluorescence during the unfolding dwell (Fig. 7B, Supplementary Fig. 24A). In control measurements with immobilized LD555-labeled proteasomes we determined the fluorescence lifetime of the LD555 donor dye as $\tau_{bleach}^{LD555} = 127$ s (Supplementary Fig. 24B), which is much longer than most of the pre-unfolding dwell times and thus confirms that the abrupt termination of pre-unfolding dwells was indeed due to substrate release rather than photobleaching of the donor dye. Wild-type proteasome released the titin I27 substrate in only $18.1 \pm 2.9\%$ of degradation attempts, and the YA1, YA2, YA3, and YA5 mutants were equally efficient in substrate unfolding (Supplementary Fig. 23D, Supplementary Table 15). Importantly, however, YA4 and YA6-mutant proteasomes showed ~2-fold higher release frequencies, letting go of $37.5 \pm 9.8\%$ and $33.3 \pm 5.1\%$ of their substrates (Fig. 7B).

Furthermore, we analyzed how long different proteasome variants tried to unfold the titin I27 substrate before either succeeding or releasing it. We found that it takes wild-type, YA6, and YA4-mutant proteasomes overall similarly long to successfully unfold titin I27 (Fig. 6C). While the YA4 mutant showed a similar time distribution for unsuccessful attempts, with many releases occurring already shortly after entering the unfolding phase, the YA6-mutant proteasome spent ~2 times longer on unsuccessful unfolding attempts before release (Fig. 7C). These ultimately unsuccessful attempts are therefore responsible for the apparently longer unfolding times of the YA6 mutant (Fig. 6H). Overall, we can conclude that the substrate-unfolding defects of YA6- and YA4-mutant proteasomes primarily originate from their significantly increased release frequency. The YA6 mutant thereby spends more time on unfolding attempts, whereas the YA4 mutant more readily releases the titin I27 substrate, suggesting an overall lower grip during unfolding power strokes. Importantly, substrate release was less frequently observed for the YA4-mutant proteasome when degrading the thermodynamically more labile titin I27$^{V15P}$ substrate (Supplementary Fig. 23E, Supplementary Table 15). It is worth noting that for wild-type, YA6-, and YA4-mutant proteasomes the time residing in non-s1 processing states before transiently switching to the s1 state, likely due to a slip during unfolding attempts (Fig. 5J, Supplementary Fig. 17), is much shorter than the average total time spent on unfolding (Fig. 7C), suggesting that a single slip does not lead to substrate release. Especially for tough-to-unfold substrates, the pore-1 loops may act like a clutch to allow slips and prevent stalling of the ATPase motor in front of an unfolding barrier, while still maintaining the substrate in the channel for additional unfolding attempts.

Unsuccessful unfolding and release of the wild-type titin I27 substrate by YA4 and YA6-mutant proteasomes is expected to cause some accumulation of deubiquitinated substrate species, either at full length or with a slightly truncated initiation region. Indeed, SDS–PAGE analyses of the end points from multiple-turnover degradation reactions indicate that, compared to the wild-type proteasome, the YA4 and YA6 mutants cause a slightly more rapid disappearance of the ubiquitinated substrate, yet increased accumulation of deubiquitinated and truncated products, which is in agreement with more frequent unsuccessful unfolding and substrate release (Supplementary Fig. 24C).

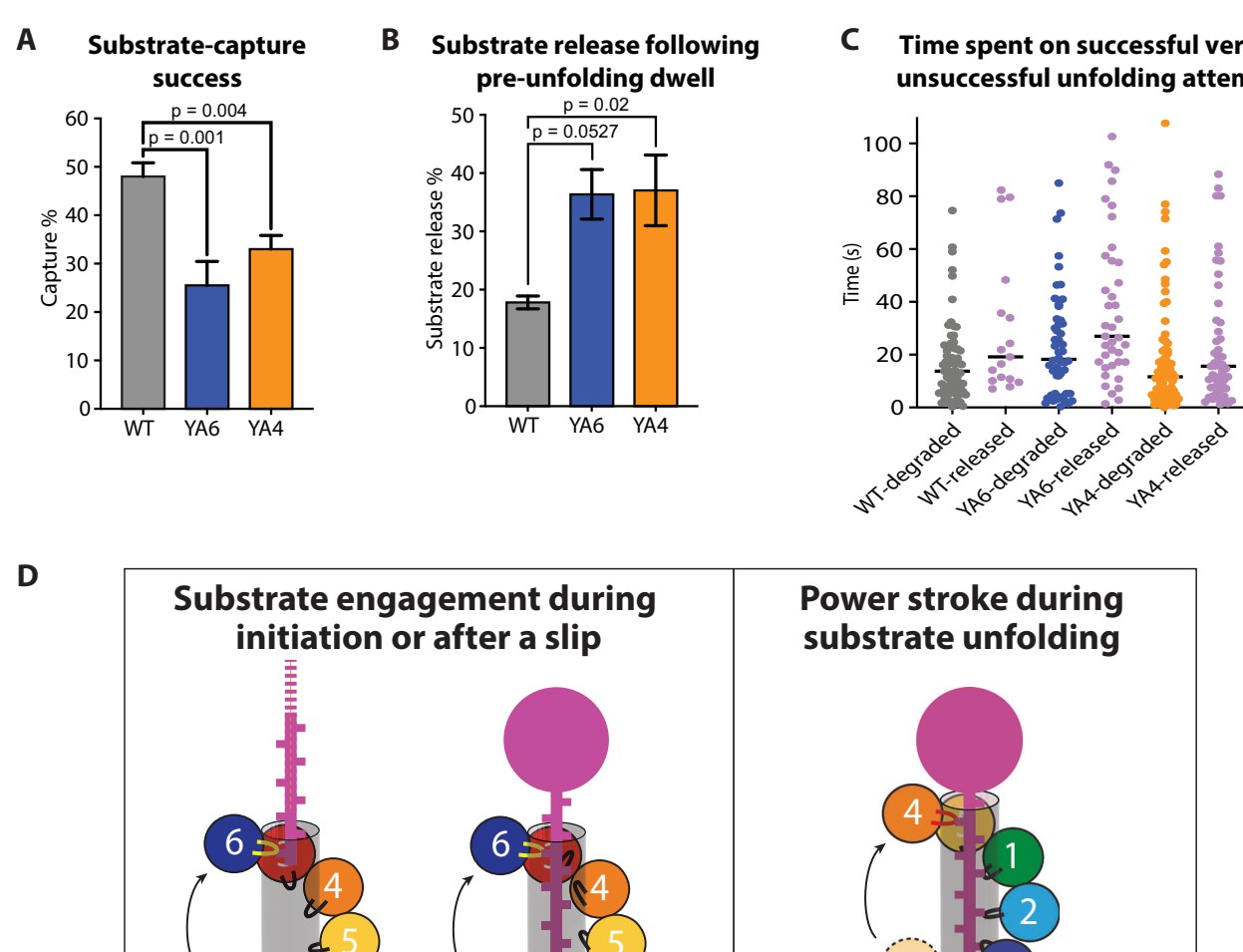

**Fig. 7 | The pore-1 loops of Rpt6 and Rpt4 are critical for efficient substrate capture and unfolding without release.** **A** Success rates for the capture of titin I27 substrate by wild-type (gray), YA6 (blue), and YA4 mutant proteasome (orange), with $n = 3$ technical replicates for all proteasome variants contributing to $N > 270$ events and error bars representing the SEM. Statistical significance was calculated using a one-way ANOVA test. **B** Percentage of titin I27 release during the unfolding dwell as determined by single-molecule FRET for wild-type (gray), YA6 (blue), and YA4 mutant proteasome (orange), with $n = 3$ technical replicates contributing to $N > 130$ events. Error bars representing the SEM. Statistical significance was calculated using a one-way ANOVA test. **C** Distributions of time spent by wild-type proteasome (gray), the YA6 mutant (blue), and the YA4 mutant (orange) on successful substrate unfolding attempts versus unsuccessful attempts that were terminated by substrate release during the unfolding dwell (magenta), as determined by single-molecule FRET ($N = 93$, 91, and 91 total events for wild-type, YA6, and YA4 mutant proteasomes, respectively). **D** Model for the critical roles of Rpt6 and Rpt4 in substrate engagement and processing. Left and middle: Rpt6 is always at the seam position in the s1-state spiral staircase of Rpts, and therefore the first subunit to either engage a substrate during degradation initiation or re-engage a substrate during processing after slipping caused transient pore-loop disengagement and a brief return to the s1 state. Right: The strongly dominant non-s1 state for the substrate-degrading proteasome has Rpt4 in the seam position of the spiral staircase. Rpt4 is therefore the first subunit to grab the substrate at the top of the staircase during an unfolding power stroke, which may be followed by rapid firing of the other Rpt subunits in a hand-over-hand mechanism.

## Discussion

Here, we uncovered that two pore-1 loops in the proteasomal heterohexameric AAA+ ATPase motor play specific roles for substrate engagement, unfolding, and the conformational response of the 19S RP. By measuring the proteasome conformational dynamics and directly observing the individual steps of substrate processing, we found that the pore-1 loop of Rpt6 is especially critical for substrate capture and stable engagement, as well as controlling the transitions between engagement-competent s1 and processing-competent non-s1 states of the 26S proteasome. Removal of Rpt6's pore-1 loop

tyrosine shifts the conformational equilibrium in the absence of substrate toward processing-competent non-s1 states. Due to the obstruction of the channel entrance by the centrally localized Rpn11 deubiquitinase in non-s1 states, this conformational shift may partially interfere with substrate insertion and, together with the reduced grip caused by the missing pore-loop Tyr, lower the success rate of capturing ubiquitinated substrates for degradation[17,38]. These findings indicate that Rpt6's pore-1 loop stabilizes the engagement-competent s1 state in the absence of substrate and/or facilitates the transition to non-s1 states upon motor engagement of a substrate polypeptide. Our cryo-EM structural studies of the substrate-free proteasome are consistent with this model, as they reveal that Rpt6's pore-1 loop adopts a helical conformation that is stabilized through interactions with the Arg fingers of the neighboring Rpt3 subunit. In addition, the pore-1 loop tyrosines of Rpt1, Rpt2, and Rtp3 are stabilized through hydrogen bonds with Rpt3, likely holding the motor in a static s1 conformation that is inactive in ATP hydrolysis. The importance of Rpt6's pore-1 loop for stable substrate engagement can be explained by its position in the spiral-staircase arrangement of Rpt subunits in the engagement-competent s1 state. Only a single staircase register has been observed for the substrate-free, s1-state proteasome, with Rpt6 in an intermediate "seam" position of the staircase[21–29,31–37]. Rpt6 is therefore expected to always be the first subunit to move up to the top of the staircase, engage the inserted substrate polypeptide, and drive the commitment step, in which the 19S RP transitions from the resting, s1 state to the processing, non-s1 states, and the system switches from passive substrate diffusion to active, ATP-hydrolysis-driven translocation (Fig. 7D). More frequent spontaneous switching to the non-s1 states combined with compromised pore-loop interactions during the first step of engaging a newly inserted substrate would explain the substrate capture and engagement defects that we detected for the YA6 mutant in our single-molecule and bulk measurements.

The importance of Rpt6's pore-1 loop for substrate engagement also explains the unfolding defects observed for the YA6 mutant. Our previous single-molecule studies revealed that during unfolding of tough substrates, the proteasome transiently switches from non-s1 states back to the s1 state, possibly due to pore-loop slippage and brief disengagement from the substrate polypeptide in the central channel[38]. It is conceivable that subsequent transitioning back to non-s1 states for continued unfolding attempts and translocation involves pore-loop re-engagement, in which Rpt6 as the seam subunit in the s1-state spiral plays again a particularly important role (Fig. 7D). Failure to rapidly re-engage the substrate would lead to delays in unfolding and eventually to substrate escape from the central channel, as we observed for the YA6 mutant.

While the YA6-mutant proteasome appears to have problems re-engaging a substrate and returning to non-s1 processing states after a slip, the YA4 mutation seems to cause more frequent slips and s1-state transitions that interrupt substrate unfolding, suggesting a reduced grip during the power stroke. This hypothesis is well supported by findings from several cryo-EM structural studies of substrate-engaged human and yeast 26S proteasomes, which consistently showed the vast majority of particles (~60–80%, Supplementary Table 16) in spiral staircase orientations with Rpt4 at either the bottom or the seam position and occupied with ADP or no nucleotide[31,37,45]. This spiral-staircase state is therefore 4–5 fold more prevalent than expected if each spiral register was equally likely to be observed during a hand-over-hand translocation mechanism. For the cryo-EM structure determinations, proteasomal substrate translocation was transiently stalled either on an uncleaved ubiquitin chain, a stably folded substrate domain, or upon addition of the non-hydrolysable ATP analog ATPγS, and it is possible that the dominant staircase orientation with Rpt4 at the bottom or seam positions represents a pre-stroke state when the motor is dealing with a tough barrier. In this staircase register, Rpt4 would be the next subunit to move up, contact the substrate

polypeptide at the top of the Rpt spiral, and drive the subsequent power stroke (Fig. 7D). Lack of Rpt4's pore-1 loop tyrosine may therefore reduce the grip during unfolding attempts, cause more frequent motor slippage, and substrate release, consistent with our experimental data for the YA4 mutant.

For distinct reasons, YA6 and YA4-mutant proteasomes both reduce the substrate-capture success by about 2-fold and increase substrate release during unfolding attempts by another factor of 2, leading to overall major defects in substrate processing. In contrast, removing the pore-1 loop tyrosines from any of the other four Rpt subunits has only minor effects on degradation. Our results, therefore, indicate asymmetric mechanisms for the proteasomal AAA+ motor in which subunits at the bottom or seam positions of specific spiral-staircase arrangements in the ATPase ring play critical roles for substrate capture and robust unfolding without release. Based on these biochemical findings in combination with previous cryo-EM structural data[31,37,45], we propose an adaptation to the current hand-over-hand translocation mechanism. Rather than uniformly progressing through the ATP-hydrolysis cycle and the different registers of the spiral staircase, the Rpt subunits may adopt a particular staircase orientation when encountering an unfolding barrier, with Rpt4 at the bottom or seam position. Coordinated, sequential ATP hydrolysis events and conformational changes may then rapidly progress around the ring until Rpt4 is in the bottom or seam position again. Such firing in bursts would explain the strongly skewed distribution of spiral staircase states in the cryo-EM structural studies of substrate-engaged 26S proteasomes, including our recent analyses of the human proteasome during active substrate degradation[46]. These structural snapshots of the human proteasome indicate that movement of Rpt4 from the bottom to the top of the spiral staircase is the slow step during the ATPase cycle of the ring hexamer and is linked to ATP hydrolysis in two subunits, Rpt2 and Rpt1, potentially representing the power stroke for substrate unfolding. This burst-like mechanism would also be consistent with the larger, 10–40 Å increments of translocated substrate polypeptide that were measured in optical tweezing experiments during substrate degradation by the related bacterial AAA+ protease ClpXP and may originate from the coordinated firing of several subunits in rapid bursts[47,48]. Although future, more detailed and time-resolved biophysical studies will be required to confirm that the 26S proteasome and potentially other AAA+ motors use such bursts in ATP hydrolysis, our results presented here clearly reveal the importance of asymmetric mechanisms for efficient substrate engagement and unfolding by the proteasomal ATPase motor and thus provide an important advance compared to structural snapshots of the hand-over-hand mechanism.

## Methods

### Cloning, purification, and labeling of the base subcomplex

Tyrosine-to-alanine mutations in the pore-1 loop of individual Rpt subunits were introduced in pETDuet1 plasmids containing a single Rpt subunit sequence (pAM395–404) by "round the horn" cloning. Briefly, primers (Integrated DNA Technologies) were phosphorylated using T4 polynucleotide kinase (New England Biolabs (NEB), Cat# M0201L) and added to a PCR sample containing the respective plasmid and amplified by Phusion High-Fidelity DNA polymerase (NEB, Cat# M0530L). PCR products were purified, ligated, and used to transform *Escherichia coli* XL1Blue cells. After the desired mutation was confirmed by Sanger Sequencing (Quintara Biosciences), plasmids for the expression of mutant base subcomplexes (pAM371–388) were constructed by subcloning using the respective restriction enzymes (NEB; PstI (Cat# R3140S), SalI (Cat# R3138S), NotI (Cat# R3189S), BlpI (Cat# R0585S), FseI (Cat# R0588S), or PacI (Cat# R0547S)) and T7 DNA ligase (NEB, M0318S), and used to transform *E. coli* XL1-Gold Ultracompetent cells (Agilent, Cat# 200314). Plasmid sequences were confirmed using whole plasmid next-generation sequencing (Plasmidsaurus). The base

subcomplex plasmids pAM431 and pAM432 were generated by "round the horn" cloning using Q5 High-Fidelity DNA polymerase (NEB, Cat# M0492S) to add a single nucleotide substitution directly in pCOLADuet-1 plasmids, with sequences confirmed as described above. Plasmid sequences are available as Supplementary Data 1, and plasmids are available upon request.

Recombinant base subcomplexes were expressed, purified, and labeled as previously described. For recombinant expression of the base subcomplex, *Escherichia coli* BL21-star DE3 cells were transformed with pAM81, pAM83, and the corresponding plasmid for Rpt1-6 subunits and pore-1 loop mutant expression (Supplementary Table 17). For unnatural amino acid incorporation, an additional plasmid, pAM87, was also co-transformed for expression of the AzF tRNA synthase/tRNA. Cells were cultured overnight in DYT media supplemented with 1x antibiotics (300 µg/mL ampicillin, 25 µg/mL chloramphenicol, 50 µg/mL kanamycin, and an additional 100 µg/mL spectinomycin for unnatural amino acid incorporation). The next day, cells were grown at 37 °C in 3 L of buffered terrific broth (17 mM $KH_2PO_4$ and 72 mM $K_2HPO_4$) media with 0.5x antibiotics to an optical density at 600 nm ($OD_{600nm}$) of 0.8, then induced with 1 mM isopropyl-β-D-thiogalactopyranoside (IPTG) for 5 h at 30 °C and then at 16 °C overnight. For unnatural amino acid incorporation, 2 mM AzF (Amatek Chemical) was added per liter before induction with IPTG. Cells were harvested by spinning at 3500 rpm for 20 min at room temperature and then resuspended in lysis buffer (60 mM HEPES pH 7.60, 100 mM NaCl, 100 mM KCl, 10 mM $MgCl_2$, 10% glycerol, and 20 mM imidazole) supplemented with protease inhibitors (aprotinin, leupeptin, pepstatin, and 4-(2-aminoethyl)benzenesulfonyl fluoride hydrochloride (AEBSF)), benzonase, and 2 mg/mL lysozyme.

The base subcomplex was purified in two affinity steps and one size exclusion step, with 1 mM ATP supplemented in all buffers. First, cells were lysed by sonication and centrifuged for 30 min at 15,000 rpm at 4 °C for lysate clarification. The eluant was loaded to a 5 mL HisTrap FF crude (Cytiva) column using a peristaltic pump, washed with NiA buffer (60 mM HEPES pH 7.60, 100 mM NaCl, 100 mM KCl, 10 mM $MgCl_2$, 10% glycerol, and 20 mM imidazole), and eluted with NiB buffer (60 mM HEPES pH 7.60, 100 mM NaCl, 100 mM KCl, 10 mM $MgCl_2$, 10% glycerol, and 200 mM imidazole). The eluant was then flowed on an M2 Anti-FLAG affinity resin (Sigma-Aldrich), washed with NiA buffer, and eluted with NiA buffer supplemented with 0.5 mg/mL 3xFLAG peptide. After concentrating with a 30 kDa molecular weight cut off (MWCO) concentrator (Millipore), the subcomplex was further purified by size-exclusion chromatography (SEC) using a Superose 6 increase 10/300 column (Cytiva) in GF buffer (30 mM HEPES pH 7.60, 50 mM NaCl, 50 mM KCl, 10 mM $MgCl_2$, 5% glycerol) supplemented with 500 µM tris(2-carboxyethyl)phosphine (TCEP) and 1 mM ATP. For labeling via unnatural amino acid, 150 µM 5,5-dithiobis-2-nitrobenzoic acid (DTNB, Sigma Aldrich) was added to the purified base after concentrating and incubating for 10 min on ice (to transiently modify and protect exposed Cys with TNB) before addition of 300 µM dybenzocyclooctane (DBCO)-Cy3 (Click Chemistry Tools), dibenzocyclooctyne (DBCO)-conjugated LD655 (a sulfo-Cy5 derivative from Lumidyne Technologies; the DBCO modification is custom synthesis) or DBCO-LD555 (a sulfo-Cy3 derivative from Lumidyne Technologies; the DBCO modification is custom synthesis). The reaction progressed overnight at 4 °C supplemented with ATP Regeneration Mix (creatine phosphate (VWR) and creatine kinase (Sigma Aldrich)) and was quenched the next day by incubating with 300 µM free AzF for 5 min and 5 mM DTT for 30 min at 4 °C (to remove the protective TNB from exposed Cys) before SEC. The concentration of base subcomplex was determined by Bradford with a BSA (Sigma-Aldrich) standard curve, and the labeling efficiency was determined by quantification of the dye absorbance using an Agilent UV–Vis Spectrophotometer.

## Purification and labeling of lid subcomplex

Recombinant lid subcomplexes were expressed, purified, and labeled as previously described[17,38]. For recombinant expression of the lid subcomplex, *Escherichia coli* BL21-star DE3 cells were transformed with pAM80, pAM86, and pAM85 (Supplementary Table 17). For unnatural amino acid incorporation, pAM314 was used in place of pAM85, and an additional plasmid, pAM87, was co-transformed for expression of the AzF tRNA synthase/tRNA. Cells were cultured overnight in DYT media supplemented with 1x antibiotics (300 µg/mL ampicillin, 25 µg/mL chloramphenicol, 50 µg/mL kanamycin, and an additional 100 µg/mL spectinomycin for unnatural amino acid incorporation). The next day, cells were grown at 37 °C in 3 L of buffered terrific broth media with 0.5x antibiotics to $OD_{600nm}$ of 0.8, then induced with 1 mM IPTG at 18 °C overnight. For unnatural amino acid incorporation, 2 mM AzF was added per liter before induction with IPTG. Cells were harvested by spinning at 3500 rpm for 20 min at room temperature and then resuspended in lysis buffer supplemented with protease inhibitors (aprotinin, leupeptin, pepstatin, and AEBSF), benzonase, and 2 mg/mL lysozyme.

The lid subcomplex was purified in two affinity steps and one size exclusion step. First, cells were lysed by sonication and centrifuged for 30 min at 15,000 rpm at 4 °C for lysate clarification. The eluant was loaded to a 5 mL HisTrap FF crude column using a peristaltic pump, washed with NiA buffer, and eluted with NiB buffer. The eluant was then flowed on an amylose resin (NEB), washed with NiA buffer, and eluted with NiA buffer supplemented with 10 mM maltose. After concentrating with a 30 kDa MWCO concentrator, HRV3C protease was added to the amylose eluate for overnight cleavage of the MBP domain at 4 °C. After cleavage, the subcomplex was further purified by SEC using a Superose 6 increase 10/300 column in GF buffer supplemented with 500 µM TCEP. For labeling via unnatural amino acid, the MBP cleavage reaction and fluorescent labeling were combined. For this, 150 µM DTNB was added to the concentrated amylose eluate and incubated for 10 min on ice before addition of 300 µM DBCO-LD555 dye, followed by addition of HRV3C protease. The reaction progressed overnight at 4 °C and was quenched the next day by incubating with 300 µM free AzF for 5 min and 5 mM DTT for 30 min at 4 °C before SEC. The concentration of lid subcomplex and labeling efficiency were determined by quantification of the 280 nm or dye absorbance using an Agilent UV–Vis spectrophotometer.

## Purification and biotinylation of 20S core particle

The purification of wild-type or AviTag-20S core was performed as previously described. Briefly, the yeast strain containing the desired 20S core particle construct (Supplementary Table 18) was grown in yeast extract, peptone, and dextrose at 30 °C for 3 days. The cells were pelleted, resuspended in core lysis buffer (60 mM HEPES (pH 7.6), 500 mM NaCl, 1 mM EDTA, and 0.2% NP-40), and popcorned into liquid nitrogen, then lysed in a Cryomill 6875D (SPEX SamplePrep). The yeast powder was thawed to room temperature, additional lysis buffer was added, and then clarified by centrifugation for 45 min at 15,000 rpm at 4 °C for lysate clarification. The 20S core particle was purified using anti-Flag M2 affinity resin. The eluate was concentrated using a 30 kDa MWCO concentrator, and the concentration was determined by absorbance at 280 nm. After biotinylation through incubation with 25 µM *E. coli* biotin ligase BirA and 100 µM D-biotin in the presence of 10 mM ATP and 10 mM $MgCl_2$ overnight at 4 °C, the complex was further purified by SEC with a Superose 6 Increase 10/300 column, equilibrated in GF buffer supplemented with 0.5 mM TCEP. The extent of biotinylation was determined in a gel shift assay by incubation with NeutrAvidin. For the wild-type core particle, the concentrated Anti-FLAG M2 resin eluate was directly loaded onto SEC, and the concentration was determined by quantification of the 280 nm absorbance using an Agilent UV–Vis spectrophotometer.

## Purification substrates

The titin I27$^{V15P}$ and titin I27 substrates were purified and labeled as described[17]. Briefly, *E. coli* BL21-star DE3 cells were transformed with pAM91 or pAM93 (Supplementary Table 17). Cells were cultured overnight in DYT media supplemented with 1x antibiotics (50 µg/mL kanamycin). The next day, cells were grown at 30 °C in 3 L of DYT media with 1x antibiotics to an OD$_{600nm}$ of 0.7, then induced with 1 mM IPTG at 30 °C for 3 h. Cells were harvested by spinning at 3500 rpm for 20 min at room temperature and then resuspended in chitin buffer (60 mM HEPES, pH 7.6, 150 mM NaCl, 2 mM EDTA, 5% glycerol) supplemented with protease inhibitors (aprotinin, leupeptin, pepstatin, and 4AEBSF) and benzonase. After lysis and clarification, the lysate was batch bound to a 15 mL of chitin resin (NEB) for 1 h at 4 °C. Then, the resin was washed with 50 mL of chitin buffer supplemented with proteasome inhibitors, 500 mM NaCl, and 0.2% Triton X-100. The substrates were eluted by overnight incubation with chitin elution buffer (60 mM HEPES, pH 8.5, 150 mM NaCl, 2 mM EDTA, 5% glycerol, and 50 mM DTT). The eluate was concentrated using a 10 kDa MWCO concentrator and loaded on a HiLoad 16/600 Superdex 75 pg column (Cytiva) with GF buffer for SEC.

## Labeling of substrates

Fluorophores were covalently attached to the substrate at its N-terminus for fluorescence polarization or gel-based measurements, or in a single engineered cysteine in the substrate unstructured initiation region for the ensemble or single molecule substrate processing assay. For N-terminal labeling, 100 µM substrate was incubated for 90 min at room temperature with 20 mM recombinant SortA enzyme and 500 µM FAM-containing peptide (5-carboxyfluorescein-HHHHHHLPETGG, Biomatik) in GF buffer supplemented with 1 mM DTT and 10 mM CaCl$_2$. The labeled substrate was enriched by clean up using a 1 mL HisTrap High Performance (Cytiva), followed by SEC on a Superdex 75 10/300 column (Cytiva) equilibrated with GF buffer. For cysteine labeling using maleimide chemistry, substrates were dialyzed for 3 h at room temperature into labeling buffer (30 mM HEPES, pH 7.2, 150 mM NaCl, 2 mM EDTA) using a Slide-a-Lyzer mini dialysis cup (ThermoFisher). Then, the substrate was diluted to 100 µM and incubated for 10 min at room temperature with 200 µM SulfoCy5 (for stopped-flow measurements, Click Chemistry Tools) or DBCO-LD555 (for smFRET measurements). The labeling reaction was quenched by reacting excess dye with 2 mM DTT and removing it by SEC on a Superdex 75 10/300 column (Cytiva) with GF buffer supplemented with 500 µM TCEP. The substrate concentration and labeling efficiency were determined by absorbance measurements using an Agilent UV–Vis spectrophotometer.

## Purification of Rpn10, ubiquitin, and ubiquitination machinery

Yeast Rpn10, *M. musculus* Uba1, Ubc4, Rsp5, and ubiquitin were prepared as described in ref. 30 using standard expression and purification procedures[30,49,50].

## Substrate ubiquitination

Enzymatic addition of ubiquitin chains to titin I27 substrates containing an unstructured region with a PPPYX motif and a single lysine residue was performed by incubating 10 µM substrate for 50 min at 25 °C in GF buffer with 10 mM ATP, 200 µM ubiquitin, 2.5 µM *M. musculus* E1, 2.5 µM UbcH7, and 25 µM Rsp5. The extent of substrate ubiquitination was determined by 4-12% Tris–Glycine SDS–PAGE.

## Single turnover degradation measured by fluorescence anisotropy

26S proteasomes were reconstituted to 2x concentration with limiting concentrations of 20S core particle (1 µM final) and saturating concentration of base, lid, Rpn10 (2.5 µM each) in GF buffer supplemented with 0.5 mg/mL BSA, 0.5 mM TCEP, 5 mM ATP, and 1x ATP Regeneration System (creatine phosphokinase and creatine phosphate) for 3 min at room temperature, followed by a 3 min incubation at 30 °C. Ubiquitinated substrate was prepared to 2x concentration (100 nM final) by diluting in GF buffer supplemented with 0.5 mg/mL BSA and 500 µM TCEP and incubated for 3 min at 30 °C. Ubiquitinated (WT or V15P) FAM-labeled titin I27 (5 µL) was rapidly mixed into reconstituted 26S proteasomes (5 µL) loaded in a 384-well clear-bottom plate (Corning). Data was acquired at 30 °C in a Synergy Neo2 Multi-Mode Plate Reader (Biotek) by following the changes in fluorescence polarization. Gain values were based on the ubiquitinated substrate alone signal. The resulting curves were fit to a double exponential decay to obtain the degradation rates. Reaction endpoint samples were run on a 4–12% Tris–Glycine SDS–PAGE gel to visualize degradation products.

## ATPase assay

ATP-hydrolysis rates were determined using an NADH-coupled assay as described previously[11]. Briefly, 26S proteasomes were reconstituted to 2x concentration under base-limiting conditions with 200 nM base, 800 nM core, 1.2 µM lid, and 1.5 µM Rpn10 in GF buffer supplemented with 0.5 mg/mL BSA, 0.5 mM TCEP, 5 mM ATP, and 0.5 mM TCEP at room temperature for 3 min. Then, 2x ATPase mix (final concentrations: 1 mM NADH, 2.5 mM ATP, 7.5 mM phosphoenolpyruvate, 3 U/mL pyruvate kinase, and 3 U/mL lactate dehydrogenase) was rapidly mixed with the proteasomes. The samples were transferred to a 384-well clear-bottom plate (Corning) and centrifuged at 1000×*g* for 1 min prior to measurement. Steady-state depletion of NADH was assessed by measuring the absorbance at 340 nm in a Synergy Neo2 Multi-Mode Plate Reader (Biotek).

## Ensemble tail insertion measurements

26S proteasomes were reconstituted to 2x concentration with limiting concentrations of SulfoCy3-labeled base (100 nM final) and saturating concentration of core particle (300 nM final), lid (500 nM final), Rpn10 (600 nM final) for in GF buffer with 0.5 mg/mL BSA, 500 µM TCEP, 5 mM ATP, and 1x ATP Regeneration System (creatine phosphokinase and creatine phosphate) for 3 min at room temperature, followed by a 3 min incubation with oPA (3 mM final) at room temperature. SulfoCy5-labeled ubiquitinated titin I27$^{V15P}$ was prepared to 2x concentration (2 µM final) by diluting in GF buffer with 0.5 mg/mL BSA and 500 µM TCEP. Data were acquired at 25 °C using an Auto SF-120 stopped flow instrument (Kintek Corporation) by rapidly mixing 26S proteasomes and ubiquitinated substrate. An excitation wavelength of 532 nM was used, and the Cy3- and Cy5-emission channels were monitored simultaneously. Gain values were determined by mixing ubiquitinated substrate with the WT base subcomplex, where no FRET is expected, but the fluorescence emission of the donor can be tracked. Since there were limiting amounts of FRET-donor (base) and excess of the FRET-acceptor (substrate), the fast phase of the Cy3 fluorescence quenching was fit to a single exponential decay function using Prism9 to obtain the amplitude and tail insertion time (Tau, s). The curves and fits were normalized to the initial fluorescence for plotting.

## Microscope set up

Single-molecule data acquisition was performed at room temperature using an inverted microscope Nikon Eclipse Ti2 (Supplementary Fig. 2), built on an ultrastable optical table. The donor dye LD555 and acceptor dye LD655 were imaged using a Nikon LUN-F laser box equipped with 532 and 640 nm lasers that were operated at ~20 mW. Power was adjusted to be 8 and 5 mW, respectively, at the sample stage. We used a filter cube from Chroma (TRF59907) placed on the Eclipse Ti2 for imaging. A ×60 oil objective (Nikon) with a 1.49 numerical aperture was used for TIRF, which we utilized with Cargille DF immersion oil and a perfect focus system (PFS) to provide active feedback stabilization. The emission from donor and acceptor dyes

was separated using image splitting optics (W-View Gemini from Hamamatsu) equipped with a T640lpxr-UF2 dichroic mirror (Chroma) and ET595/50-m and ET655lp bandpass filters (Chroma) and simultaneously imaged with an EMCCD camera (Andor Technology, iXon Ultra 512 ×512 DU-897) cooled to −70 °C.

For registration of the two emission channels, images were acquired with 100 nm TetraSpeck beads (Thermo Fisher Scientific). Reaction chambers were assembled on microscope slides using double-sided Scotch tape and glass coverslips, and a 1:10 dilution of TetraSpeck beads in GF buffer was flowed into the chamber. The dilution was incubated for 2 min, the excess beads were washed out using the GF buffer, and the sample was imaged using the 532 nm laser. Registration was then performed by overlaying cropped regions from the donor and acceptor emission channels such that the bead positions in each channel agreed. The data acquisition for TetraSpeck beads was performed with the following camera settings: EM gain at 4, readout rate at 17 MHz, and preamp set to 1, with activated overlap mode and camera internally triggered through the NIS-Elements software (Nikon).

### Single-molecule fluorescence microscopy data acquisition

Data were acquired with slight modifications to our previous protocol[38]. Briefly, reaction chambers were assembled on glass microscope slides using double-sided tape and polyethylene glycol (PEG)-coated coverslips with low-density PEG-biotin (MicroSurfaces Inc.). The reaction chambers were incubated with NeutrAvidin (0.25 mg/mL; Thermo Fisher Scientific), and excess NeutrAvidin was removed by washing with 40 μL of assay buffer (GF buffer supplemented with 0.3 μM Rpn10, 1.2 mM Trolox, 0.2 mM β-mercaptoethanol, 0.5 mg/mL BSA, 2 mM ATP, and an ATP regeneration system (creatine kinase 0.03 mg/mL and 16 mM creatine phosphate)). Proteasomes were reconstituted by incubating 500 nM biotinylated core, 400 nM base, 600 nM lid, and 1 μM Rpn10 in assay buffer supplemented with 0.5 μM TCEP for 10 min at room temperature. Reconstituted proteasomes were flowed into the reaction chamber at 1 nM and incubated for 5 min. Excess proteasomes in solution were washed out with 40 μL of assay buffer, followed by 20 μL of imaging buffer (assay buffer supplemented with the protocatechuic acid (PCA, Sigma Aldrich)/protocatechuate 3,4-dioxygenase (PCD, Sigma Aldrich) oxygen scavenging system). For the conformational change assay, 0.5 or 1 μM unlabeled ubiquitinated substrate was added to the imaging buffer. For the ATPγS samples in the conformational change assay, proteasomes were washed after the 5 min incubation with assay buffer containing 2 mM ATPγS in place of ATP and the ATP Regeneration system, followed by imaging buffer containing the same substitution. For the substrate-processing assay, 100 nM LD555-labeled substrate was added to the imaging buffer.

For all single-molecule acquisitions, the acceptor fluorophores were imaged first by taking a single-frame snapshot after a 640 nm laser excitation, which we used to evaluate the extent of surface deposition and localize particles labeled with the LD655 dye. Then, the donor and acceptor fluorescence signals were simultaneously monitored following 532 nm laser excitation. The acquisition rate for these measurements was 20 Hz, and the acquisition settings for proteasome samples were done using the same parameters as for TetraSpeck beads described above, but with a higher EM gain of 40. The degradation activity of reconstituted, dye- and biotin-labeled proteasomes was confirmed in bulk assays as previously described[17,38].

### Single-molecule substrate processing assay data analysis

We analyzed only singly capped and singly labeled proteasomes, which were identified based on their total fluorescence and a single photobleaching step. Data were analyzed with slight modifications compared to our previous study[38]. Briefly, raw images were aligned using a custom MATLAB script (https://github.com/jabard89/tirfexplorer)

and loaded on Fiji to parse using the plugin Spot Intensity Analysis (https://imagej.net/plugins/spot-intensity-analysis) to select particles. The generated traces were then sorted and analyzed using a custom-built MATLAB app as previously described[38]. The start and end of each substrate-processing event were determined from the total fluorescence, which depends on the residence time of a donor-labeled substrate, and the apparent FRET efficiency values outside substrate-processing events were not calculated (indicated by dashed lines in all related figures) for easier scoring of substrate-processing events. Substrate processing traces were inspected manually and scored for various features to distinguish the processing steps of degradation. The tail insertion phase was measured from the appearance of a donor signal and intermediate apparent FRET efficiency (~0.50) to the time of the first high-FRET efficiency peak. Deubiquitination was measured as the phase between the first high-FRET efficiency peak and the last peak before donor recovery (FRET-signal decay). The time for the pre-unfolding dwell was measured by measuring the time spent at ~0.40 FRET efficiency after tail insertion or deubiquitination, until decay from ~0.20 FRET efficiency. The translocation phase was scored as starting at the FRET-efficiency signal decay to ~0.20 and with the persistence of the donor signal, and ending when the donor signal was lost. The values obtained for each scored step were plotted as one-cumulative distribution functions (1-CDF) using Prism9.

To determine the substrate-capture success, the fractional number of productive binding events leading to degradation relative to the total number of substrate encounters (both productive and non-productive) was calculated. Since substrates were donor-labeled and proteasomes acceptor-labeled, we considered only substrate binding events where an increase in donor fluorescence colocalized with acceptor fluorescence, leading to an apparent FRET efficiency value of 0.50 upon substrate binding. Binding events were scored as productive and indicative of degradation if they showed a high-FRET phase followed by an extended donor recovery and fluorescence dwell as a measure of successful threading after unfolding. Binding events were scored as nonproductive if the ~0.50 FRET efficiency was not followed by an increase to high–FRET efficiency or recovery of the donor fluorescence. A minimum of eight movies, containing at least 22 binding events per movie, were analyzed for each condition, and the error is the SD for the total number of technical replicates.

To determine the substrate-release following the pre-unfolding state in substrate-processing assays, we calculated the fractional number of events where a persistent pre-unfolding dwell was observed relative to the number of events where degradation was completed (productive events). Substrate processing was evaluated as complete if there was a decrease in the FRET efficiency from ~0.40 to ~0.20 (signifying a progression from the pre-unfolding dwell to translocation of the unfolded protein) and then persisting at ~0.20 during translocation. Substrate release was evaluated as the abrupt loss of donor and acceptor signal during the unfolding dwell, i.e., at the ~0.40 FRET-efficiency state, with no decrease in FRET signal from ~0.40 to ~0.20 or a persistent 0.20 FRET efficiency. As a control to account for dye photobleaching, the LD555 donor survival was determined by assembling 26S proteasomes with LD555-labeled base subcomplex, which were prepared and imaged as described above. The survival of the LD555 emission was calculated by measuring the time of photobleaching for single-labeled particles. A total of 152 particles were measured, which yielded 113 photobleaching events that were plotted in a 1-CDF plot.

### Data analysis for single-molecule conformational dynamics assay

In addition to ebFRET, which was previously used to analyze proteasome conformational dynamics[38], a BNP-FRET method was employed. For the ebFRET analysis, raw movies were first aligned using a custom script as described above and parsed using the software package tools.

Traces were sorted, plotted with Spartan software's built-in functions, and analyzed using hidden Markov modeling with ebFRET[51]. The dwell time distributions for each FRET state were extracted and best fit to a double exponential using Prism9 to obtain transition rates. FRET efficiency distribution plots were also produced using Spartan software's built-in tools.

BNP-FRET, on the other hand, does not require any preprocessing of traces and directly incorporates an accurate image formation model to estimate probability distributions and uncertainty over all parameters of interest, including the number of states, transition rates, and FRET efficiencies. Furthermore, to ensure trace-to-trace accuracy of estimates, we calibrated bleed-through noise and EMCCD settings (EM gain 40, readout rate 17 MHz, preamp 1) and pixel-to-pixel noise parameters (dark counts and bias/offset). Particles were selected based on the presence of both donor and acceptor fluorescence, and traces were generated and filtered to obtain signals from proteasomes that were singly capped, singly donor and acceptor-labeled, and showed transitions between FRET-states at rates >0.2 s⁻¹. To calculate the conformational transition rates for proteasomes in the absence and presence of substrate, BNP-FRET uses MCMC sampling techniques, which implement a Bayesian non-parametric framework (Supplementary Fig. 4B) that does not pre-determine variables like the number of proteasome states. For each smFRET time trace, over 20,000 samples for state trajectory, rates, and FRET efficiencies were generated and pooled together in an ensemble-level bivariate probability distribution and transition-density plots. The rate values, standard deviation, and weights for the high and low FRET states were selected by fitting histograms derived from the bivariate distribution plot using a multivariate Gaussian mixture model. To determine a baseline of "stuck" or transition-less proteasome particles, individual traces from the dataset for wild-type proteasome in ATP having mean Gaussian values rates <0.2 s⁻¹ were removed from further analysis in all datasets. The scripts for kinetic analysis are available through a GitHub repository at https://github.com/LabPresse/BNP-FRET-Binned.

Lastly, FRET efficiency distribution plots were generated using the Spartan software, as previously described[38]. Substrate processing dwell times were scored manually from the onset to the stable end of high apparent FRET efficiency, including brief excursions to low-FRET efficiency during the processing of the titin I27[V15P] substrate.

## Multiple turnover degradation measured on SDS–PAGE

Proteasomes were reconstituted at 2X concentration with limiting concentrations of 20S core particle with 100 nM core, 400 nM base, 800 nM lid, and 1 μM Rpn10 in GF buffer supplemented with 0.5 mg/mL BSA, 0.5 mM TCEP, 5 mM ATP, and 1x ATP Regeneration Mix (creatine kinase and creatine phosphate) for 3 min at room temperature. The FAM-labeled titin I27 substrate was prepared at 2X concentration (4 μM) in GF buffer. Gel-based multiple-turnover measurements of FAM-titin I27 degradation were initiated by the addition of 5 μL of ubiquitinated substrate to 5 μL of proteasome sample. Aliquots (1.2 μL) were taken after 60 min and quenched in 2X SDS–PAGE loading buffer (5 μL). The reaction samples were run on 16.5% Tris–Tricine SDS–PAGE gels (Bio-Rad) and imaged on a Typhoon variable mode scanner (GE Healthcare) for fluorescein fluorescence.

## Cryo-EM sample preparation, data collection, processing, and model building

Rpn11-FLAG-tagged *S.c.* 26S proteasomes purified from yeast were applied directly to glow-discharged (25 mA, 25 s) UltrAuFoil® R 2/2 200 Mesh Au grids (Q250AR2A, Electron Microscopy Sciences). The proteasome concentration was at 2 μM in 60 mM HEPES, pH7.4, 25 mM NaCl, 25 mM KCl, 10 mM MgCl₂, 5 mM ATP, 2.5% glycerol, and 0.02% NP-40. Samples were blotted for 3 s and plunge frozen in liquid ethane using a Vitrobot (ThermoFisher) at 100% humidity and 20 °C. Clipped grids were transferred to a Talos Arctica 200 kV electron microscope

with a Gatan K3 camera. A total of 2999 movies were collected using SerielEM[52] in super resolution mode with 0.5575 Å per pixel, a defocus range of −0.5 to −1.5 μm, and a total electron dose of ~50e Å⁻².

CryoSPARC v4.4.0 was used for all data processing. Movies were processed on the fly with CryoSPARC live, in which patch motion, patch CTF corrections, blob picker (200 Å min and 600 Å max dimensions for particles) were used, followed by extraction with a 660 box size. All particles were subject to rounds of 2D classification before generating 2D templates; 5 views were chosen for template picking. A total of 172,186 particles were extracted before 2D classification by ab initio reconstruction. A 30S proteasome model was chosen and seeded 4 times in heterorefinement to remove junk particles. One major class emerged with 92,861 particles of the 30S proteasome (potentially a mix of 26S and 30S proteasomes), which was subject to homorefinement with C2 symmetry applied. Particles were then symmetry expanded in C2 and realigned on the RP before re-extraction with a 360 box size. This method effectively doubles the number of particles and focuses on one end of the 30S. 3D reconstruction was used to generate a 19S RP model from previous alignments and to seed 3 classes in heterorefinement. Heterorefinement emerged with 3 classes labeled as junk (due to lots of noise and smeary density), non-s1-state particles, and s1-state particles. Non-s1-state particles were refined using homorefinement, but the resolution was limited, likely due to a mix of conformational states and the low number of particles, and the data were not processed further. We subjected s1-state particles to local refinement with a mask around the 19S RP with non-uniform refinement and per particle scale factors on.

While the s1-state map was at moderate resolution (~3.6 Å), the sharpened map showed high-resolution features that allowed side-chain modeling. Model building used the structure of the ubiquitin-bound yeast proteasome (PDB ID: 6J2Q) as a starting model, where subunits are roughly fit using ChimeraX and fit-to-map. Models were extensively rebuilt using Coot[53], and PHENIX[54,55] was employed for docking and rebuilding of AlphaFold models[56]. The model and sharpened map were refined with PHENIX real-space refinement, and local resolution was estimated in CryoSPARC[57]. Figures were generated using PyMOL (The Molecular Graphics System, Version 1.8, Schrödinger, LLC; http://www.pymol.org/), UCSF Chimera, and ChimeraX.

### Reporting summary

Further information on research design is available in the Nature Portfolio Reporting Summary linked to this article.

## Data availability

All data generated or analyzed during this study are included in this manuscript and the Supplementary Materials. Source data are provided with this paper. Structural data are available in the Electron Microscopy Databank and the RCSB Protein Databank (EMDB ID 45579 and PDB ID 9CGC for the non-processing, s1-state 26S proteasome). For additional information and requests for resources, reagents, and constructs, please contact the lead author, Andreas Martin (a.martin@berkeley.edu). Source data are provided with this paper.

## Code availability

A custom MATLAB script was used to align raw TIRF images and is available at https://github.com/jabard89/tirfexplorer. The scripts for the BNP-FRET kinetic analysis are available at https://github.com/LabPresse/BNP-FRET-Binned.

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

## Acknowledgements

We thank Zaw Htet and Ken Dong for their help in data analysis, and all members of the Martin lab for their discussion and support. This research was funded by the Howard Hughes Medical Institute (C.A., C.L.G., and A.M.) and by the US National Institutes of Health (R01-GM094497 to A.M.). S.P. acknowledges support from the US National Institutes of Health Grants R01-GM134426, R01-GM130745, and MIRA R35-GM148237. E.L.A acknowledges support from the National Academies of Sciences, Engineering, and Medicine Ford Foundation Pre-Doctoral Fellowship.

## Author contributions

E.L.A. and A.M. conceived the study and designed experiments. E.L.A. cloned constructs, expressed, purified proteins, and performed biochemical measurements as well as data analyses. E.L.A., S.Z., and H.H. performed and analyzed single-molecule experiments. A.S. and S.P. performed BNP-FRET analyses of single-molecule data. C.A. performed cryo-EM sample preparation, data collection, and data processing. C.L.G. and C.A. generated atomic models based on cryo-EM maps. E.L.A. and A.M. wrote the manuscript with comments from all authors.

## Competing interests

The authors declare no competing interests.
