## [Transparent Peer Review file · Nature Communications]

Substrate-interacting pore loops of two ATPase subunits determine the degradation efficiency of the 26S proteasome

Corresponding Author: Professor Andreas Martin

Version 1:

Reviewer comments:

Reviewer #1

(Remarks to the Author)

The paper by Lopez-Alfonzo et al describes new mechanistic insights into the unfolding and degradation of proteins by the proteasome. Upon recognition of ubiquitinated proteins, the pore loops of the proteasomal AAA-ATPase ring engage with the substrate's initiation region thereby facilitating the unfolding and translocation of substrates into the CP. The authors identify the loop of Rpt6 to be critical in initiation. Interestingly, in the engagement competent state (s1) this loop interacts with Rpt3 arginine fingers, which contributes to stabilizing this s1 state. So, the Rpt6 loop is a critical component of the initiation of degradation. A second loop that was identified as important is the Rpt4 loop, however, the logic and mechanistic unique feature here remains less well defined. The authors also observed some unique properties for Rpt1 loop mutants, however, no rational or detailed follow up for that was provided.

Overall, the strengths of the paper are the elegant molecular dissection with powerful biophysical assays such as the 26S conformational state with a single molecule FRET-based assay and the substrate translocation with another FRET based assay. These assays have been used in the past and have been further improved upon in this manuscript. These assays are further supported with a cryoEM structure that is of higher resolution as previous structures. This was important to identify the Rpt6 pore loop-Rpt3 Arg finger interactions.

The manuscript is well written and provides strong support for the proposed models and compelling and relevant new insights into the mechanism of proteasome substrate engagement and unfolding. One weakness is that the Rpt1 loop mutant data are largely not explained and show in part phenotypes similar to the Rpt6 loop mutations, suggesting that some of the data can be interpreted differently as Rpt1 loop clearly must be acting differently from the Rpt6 loop.

Another point of concern is that the general phenomenon that are deduced from these in vitro studies are all based on one model substrate. While the structure of the engagement competent 26S supports the Rpt6 role as gate keeper and thus likely is substrate independent, for some of the other loops that is less clear to me. E.g., the described important role of Rpt4; could there be substrate dependence or other differences where certain loops might play a more dominant role in one case compared to another? Different initiation regions have different efficiencies of degradation, so maybe the length and/or composition of the tail can show more dependence on different loops? So, how can the author be sure the data for Fig. 1D, e.g., is more universal and not specific for this Titin substrate? I think it would be helpful if the authors could explain why this is universal or state the constraints of their study in this regard.

The Rpt1 mutant shows unique and interesting ATPase activity with and without substrate. It is not clear to me how does this mutant fits with the authors' model? Rpt1 mutant has increased ATPase activity for 26S by itself (like Rpt6 loop mutant) without an shift in the population of s1 to non-s1. So, what is going on with Rpt1 mutant and how can authors be sure their interpretation for Rpt6 mutant is correct as there can apparently be different causes of increased ATPase activity? If the higher rate of Rpt1 mutant is for increased activity in the non-S1 state, then combining Rpt1 and Rpt6 mutation should be additive regarding ATPase activity and help distinguish in mechanism between these mutants on 26S by itself? Line 274-275. I like this conclusion by the authors. However, doesn't the Rpt1 mutant suggest this might be too simplistic, as there is higher activity without increased non-s1 state?

Mutating the loop of Rpt6 leads to increase non-s1 conformations, which are less suitable for engagement with initiation region. Based on cryoEM the loop of Rpt6 interacts with Rpt3 arginine fingers. The complementary mutations of Rpt3 should thus also change the equilibrium from more s1 to more non-s1, which could be a nice confirmation of their data, assuming the mutated arginine fingers don't interfere too much with ATP binding and ATPase ring formation. Did the authors attempt

or consider this mutation?

The supplementary Fig 7-2 C provides for nice complementary data to the other type of assays. However, looking at the figure the impact seems rather modest compared to the described impact of the mutations from the other assays. Wouldn't it be expected that more of the substrate would be degraded in the wild type versus Rpt6 and Rpt4 mutants, this is not apparent to me from this assay. Also, would all the released substrates be truncated or could some be full length and only be deubiquitinated?

Reviewer #2

(Remarks to the Author)

This manuscript by López-Alfonzo, Erika M., et al., presents a comprehensive study using single-molecule FRET and cryo-EM techniques to elucidate the role of pore loops in the Rpt6 and Rpt4 ATPase subunits of the 26S proteasome. The authors provide valuable insights into how these structural elements affect the protein degradation process, demonstrating that the Rpt6 pore loop is crucial for substrate handling and initiation of degradation, whereas the Rpt4 loop aids in substrate unraveling and sustains proteasome efficiency. Although the experiments are well-executed and the findings presented are intriguing, I have the following major concerns:

1. The study explores the Rpt6 pore-1 loop's role via the YA6 mutation, positing that it affects substrate capture. Could you detail whether this mutation alters the interaction between Rpt3 and Rpt6? Furthermore, how does the helical conformation of the Rpt6 pore-1 loop interact with the arginine finger of Rpt3 to facilitate substrate capture and the conformational state transition?
2. The manuscript references data on ATP_γS in Figure 2's legend that appears absent from the figure itself. Could you include a supplementary explanation or correct the figure to reflect this data?
3. You mention kinetic parameters from BNP-FRET analysis for transition rates; however, the methodology to derive these figures is unclear. The visual distinction in Figure 2D between states is not evident. Could you elaborate on the criteria used for determining these rates?
4. In Figure 2E, the fitting of k_{fast} and k_{slow} for the escape rate of the low-FRET s1 state suggests two kinetic processes. If these rates are not indicative of separate processes, please clarify their relevance.
5. The authors have excluded static fluorescence traces from the population analysis. It would be beneficial for the study if the reasons for these traces exhibiting static characteristics were discussed. Specifically, it should be clarified whether these static traces originate from denatured samples, which could potentially exhibit different fluorescence behaviors. If the static nature is not due to denaturation, an explanation should be provided regarding why these samples exhibit such characteristics. This discussion could help validate the exclusion of these traces and ensure that the analysis accurately reflects the behavior of functional protein complexes.
6. The Rpt1 mutant YA1 shows enhanced ATPase and degradation activities. Could further insights into the mechanistic implications of this observation be provided, particularly regarding the pore loop's role as a potential rate-limiting factor?
7. The role of the YA6 mutation in facilitating transitions to non-s1 states lacks direct evidence linking substrate interaction directly with Rpt6. Structural data in Figure 4 imply dependency on Rpt3. Could the mutation's effects be due to destabilizing Rpt3/Rpt6 interactions? Direct structural evidence of substrate engagement with Rpt6 would strengthen this claim.
8. The observation of increased substrate releases due to mutations in the Rpt4 pore-1 loop suggests a gripping defect. Could an additional mechanism explain this loop's role in substrate processing?
9. You propose a "burst-like" mechanism contrasting with the traditional hand-over-hand model. How does this align with the asymmetric arrangement of Rpt subunits and the structure of individual subunits? More data would bolster this novel hypothesis.
10. The use of singly fluorescently labeled substrates could potentially influence degradation rates. Have comparisons been made between rates of ATP hydrolysis and protein degradation in fluorescently labeled versus non-labeled substrates?
11. To substantiate the claim that mutants YA4 and YA6 destabilize the non-S1 state with the substrate (Figure 5J), could you provide dwell-time distributions for high-FRET states across other mutants?
12. In Figure 6, the definition of FRET states shows significant fluctuations, especially in the unfolded state (Figure 6D). Could you provide a detailed methodology for how these FRET states were determined and assigned?

Reviewer #3

(Remarks to the Author)

Using FRET-based single-molecule analysis, López-Alfonzo et al. examined six Pore-1 loop mutants, where the conserved Tyr residues in the Pore-1 loop were substituted with Ala, revealing that the Pore-1 loops of Rpt6 and Rpt4 play unique and distinct roles compared to other subunits. They developed an algorithm to improve the analysis of the previously established smFRET method, enabling unbiased assessment of structural fluctuations. Structural analysis using Cryo-EM demonstrated that the Rpt6 Pore-1 loop, whose mutation resulted in defects in both ATPase activity and substrate degradation assays, forms an α -helix in the s1 state, accompanied by a unique Arg finger orientation. Overall, the experiments were well-designed, with the structural dynamics derived from their FRET analysis aligning with those observed in Cryo-EM. The established smFRET analysis provides significant and unique insights into the mechanical function of the proteasome. While a few minor questions remain, the study is well-suited for publication in Nature Communications.

1. YA6 increases the non-s1 conformation in the presence of ATP, while no significant changes are observed in YA1 and YA4. The escape rate, representing the rate of conformational switching, is notably faster than the ATPase hydrolysis rate,

implying that the conformational switch is independent of hydrolysis. What, then, triggers the conformational switch? Additionally, how does hydrolysis influence this process?

2. In Cryo-EM analysis, it is important to show the density map to validate whether the side-chain structures are accurately modeled.

3. Among the structures of the yeast 26S proteasome, s2 and s5 exhibit similar configurations to the s1 state, with Rpt6 positioned as the seam subunit. Do these structures share loops that form a helix with Tyr222 oriented inward? Additionally, can this helix formation of the pore-1 loop be observed when other subunits occupy the seam position? A comparison would be interesting.

4. smFRET data showed that the YA4 mutant significantly increases the s1 conformation in the presence of substrate, presumably due to substrate slip. The reviewer questions whether a single mis-capture could indeed lead to entire substrate release. What is the probability that a mis-capture of the substrate causes its release? Would it be feasible to analyze the frequency of substrate release to investigate this further?

5. A more detailed discussion on the dysfunction of YA4 would be valuable, as the data plausibly suggest that Rpt4 acts as a barrier to switching motion. In Cryo-EM analyses, s1-like and s4-like conformations are frequently observed in the absence and presence of substrate, respectively. This may result from the instability of other seam configurations. However, it remains unclear how the Rpt4 loop actively facilitates the transition from the bottom position to the seam.

Reviewer #4

(Remarks to the Author)

Kinetics analysis for single-molecule FRET traces

> To analyze the FRET transition kinetics, authors used the Hidden Markov Model, a well-established model for single-molecule analysis. They used "ebFRET" software for this, and transition density plots show the transition between two dominant FRET states for the proteasome construct. They also used Bayesian nonparametric FRET-analysis algorithm based BNP-FRET which does not assume the number of states.

The authors were to better explain how transition rates were obtained and if the BNP-FRET result was different from the ebFRET result.

Comments on the transition rate analysis

They were to clearly specify the used fitting model (single or double-exponential) and overlay the fitting curve with the dwell time data in Supp. Fig.2-2 or elsewhere to show if their model actually fits. It seems that they used double exponential fitting to get the "fast" and "slow" rates.

Because we expect the single-exponential distribution (with a tail...) in the simple two-state system (assuming one dominant rate-limiting step) and double exponential fitting usually fits many types of data, we need some justification for the double exponential fitting. Is it due to the limited temporal resolution or another technical limitation? Or, is a two-step reaction actually expected here?

Comments on each line

220: with rates of 0.34 - 4.22 s⁻¹ ... 0.66 - 7.22 s⁻¹

> What does the range mean? Confidence interval? The range is not very clear or informative for a data set with an intrinsic wide distribution...

222: According to the ebFRET analysis (Fig. 2E)

> Fig. 2E legend says the data is Bivariate probability distribution (not ebFRET). Which one is correct?

> Describe what three dots represent, here. t₁/t₂ of double exponential fitting of each data set? How can we get that from Fig. 2D?

Fig. 2D/2E: The data should be ATP vs ATP_rS, but there is only one graph for each subplot.

225: The rates determined by ebFRET and BNP-FRET show overall similar trends and the observed differences ... using camera calibration frames to avoid overfitting.

> I believe the author should find a better analysis (may depend on the parameter we want to see) and provide a fair justification for using the method over the other one. If the difference is from the various noises, we cannot trust the data for the analysis.

In my opinion, because BNP-FRET does not assume the number of states, the result can be less clear. If we can confirm that there are only two states based on the traces, the ebFRET result can be more clear. BNP-FRET can be used to confirm if there are 3rd or more states.

238: This was confirmed in the BNP-FRET analysis by a redistribution of the s1-to-non-s1 escape towards higher rates that extend beyond 6 s⁻¹ (Fig. 3E..

> IT seems that both states got longer escape times in Fig. 3E, which means that the transition got slower which does not necessarily mean that the high FRET state is more populated. We'd better check the numbers or the bar graph.

The single-molecule distribution of dwell time (escape time or rate) is informative. It is hard to follow the distribution in the 2D density plot for me.

Fig3

> Labeling non-S1 or S1 on each FRET peak would help.

Fig 3G/H

> a color legend for WT/YA6/YA4 would help.

243 The more frequent sampling of processing-competent non-s1 states by the YA6 mutant in the absence of substrate is also consistent with its higher basal ATPase rate.

> As mentioned at 251, YA1 is not like this. It is not convincing... we may need a better explanation or insight on the structure.

285: We previously found that upon substrate engagement ... high-FRET non-s1 states that are maintained until substrate unfolding and threading through the ATPase

> Based on this, we will assume that high-FRET state dwell is the sum of intrinsic high FRET, substrate folding time (after binding), and threading time.

288: With the new instrumental setup

> The author mentioned this multiple times. Need to specify what was improved technically (eg. temporal resolution?) and what information authors additionally obtained. Or remove it.

289: high-FRET phases, i.e. substrate-processing dwells

> unfolding and threading time (285)

299: we were previously unable to use ebFRET for reliably measuring the ...

> I do not get the difference between ebFRET and BNP here. Both measure the dwell time of each state. It seems that the bivariate probability distribution of BNP-FRET shows the distribution of the "rate" (the inverse of "dwell time") of each event. If so, there should be no significant difference between the two methods in principle. If not, bivariate probability distribution should be better explained.

> It is not easy to compare bivariate probability distributions. In this figure. The sum of all pixels should be 1, but Fig 5I has much more bright pixels compared to 5G. It seems that they are not normalized equally or the color scale is not showing the proper region.

306: higher rates of 0.38 - 8.72 s⁻¹, compared to 0.55 - 6.61 s⁻¹ for WT

> not a good way to represent the data

309: correspondingly wider range of rates

> "wider" what does it mean? Is the rate more heterogeneous or higher?

315: To more specifically analyze how

> need to specifically explain the benefit of the manual scoring...

> Was the manual scoring result different from the ebFRET or BNP? With the slower rates, both software may find the state better, too.

320: shorter high-FRET phases, with YA4 = 3.2 ± 0.3 s and YA6 = 3.1 ± 0.3 s

> Specify the representative numbers. Are they mean?

> I suggest the author show the distribution of dwell time (frequency vs dwell-time) as many conventional single-molecule data and discuss the distribution more. It seems that WT has a more Gaussian distribution which implies that the high FRET dwell time is the result of multiple reactions, or threading. YA6 and YA4 are not. It seems that the result is a mixture of failing (single or double exponential) and some threading.

327: Rpt4's pore-1 loop affects the stable maintenance

> Specify how it was affected

Fig. 5C-E

> labeling (WT, YA6, YA4) would be nice.

353: slower substrate insertion into the YA6 mutant can be explained by its shift of the conformational equilibrium

> YA4 showed the slower insertion, but its shift of the conformational equilibrium is the opposite.

360: suggesting that substrates either do not fully enter the ATPase ring or fail to stay stably inserted

> If so, oPA does not work for mutants and we cannot assume the single-turnover scenario. Is it still fair to accept the measured time constant?

1072: Traces were filtered for visualization using Matlab R2019b.

> Specify the filtering criteria instead of the software.

More minor points

184: Phe2

> What is Phe2?

592: 5,50-dithiobis-2-nitrobenzoic acid

> 5,5-thiobis?

627: 300 uM LD555 (Lumidyne Technologies)

> DBCO? Cat No. Can be added to all reagents. Specify dyes and their functional group as possible.

680: Excess dye was quenched

> The dye was not quenched here. The reaction was.

742: built on an ultrastable single molecule stage

> There are many words for optical tables. Single-molecule stage sounds a bit weird.

746: at 5mW and 2mW power

2mW laser (from the laser head) is usually not good enough for single-molecule imaging in general. Better specify the information.

793 but with a higher EM gain of 40

> The 40 cannot be a real gain of iXon for smF...

Supp. Fig.2-1:

> The camera model number is wrong.

Supp. Fig.2-2: "hmm-mcmc"

> "hmm-mcmc" was not mentioned before. It should be a BNP-FRET-related one. Please use consistent terminology.

"DTNB was added to the purified base after concentrating"

Is it to minimize the non-specific interaction of DBCO? Does blocking thiol affect the protein function?

Q. The construct shows the transition between two states (S1 or non-S1). Do substrates bind to both states?

Version 2:

Reviewer comments:

Reviewer #1

(Remarks to the Author)

authors have addressed all my concerns and questions.

Reviewer #2

(Remarks to the Author)

The authors have addressed all of my concerns. I have no further suggestions and recommend the manuscript for publication.

Reviewer #3

(Remarks to the Author)

By responding thoughtfully to my comments and that of the other reviewers, the authors have greatly strengthened and improved the manuscript. This paper is exceptional in both its biophysical approaches and its analysis of proteasome functions, and it is fully deserving of acceptance in Nature Communications.

Reviewer #4

(Remarks to the Author)

The authors' response and the new revision are acceptable.

One remaining minor concern is related to Fig. 2E.

The statement, "Three technical replicates with separate ebFRET analyses, contributing to N = 289 particles," could be interpreted as indicating that the main figure is based on a single experiment whose dataset was divided and analyzed three times, rather than on three independent biological experiments. If this is the case, the data may not be sufficiently robust for inclusion as a main figure. If the authors indeed performed three independent experiments, it would be clearer and more appropriate to state this explicitly.

We would like to thank the reviewers for their detailed review of our manuscript and their constructive criticism that helped us improve our study with additional experiments and analyses. Our *responses to individual comments are pasted below in blue and corresponding changes to the manuscript are highlighted in red.*

REVIEWER COMMENTS

Reviewer #1 (Remarks to the Author):

The paper by Lopez-Alfonzo et al describes new mechanistic insights into the unfolding and degradation of proteins by the proteasome. Upon recognition of ubiquitinated proteins, the pore loops of the proteasomal AAA-ATPase ring engage with the substrate's initiation region thereby facilitating the unfolding and translocation of substrates into the CP. The authors identify the loop of Rpt6 to be critical in initiation. Interestingly, in the engagement competent state (s1) this loop interacts with Rpt3 arginine fingers, which contributes to stabilizing this s1 state. So, the Rpt6 loop is a critical component of the initiation of degradation. A second loop that was identified as important is the Rpt4 loop, however, the logic and mechanistic unique feature here remains less well defined.

We thank the reviewer for this positive assessment of our findings. Regarding the contributions of Rpt4, our data show that a lack its pore-1 loop Tyr leads to ~ 2-fold reduced success in capturing a substrate and a ~ 2-fold increased release of substrate due to unsuccessful unfolding attempts. Together with the strongly skewed cryo-EM particle distributions observed in numerous studies, where spiral-staircase states of the ATPase motor with Rpt4 at the bottom or the seam position are 4 to 5-fold more prevalent than expected for an equal distribution of all staircase states, these data indicate a critical role of Rpt4 in the mechanical power stroke for substrate unfolding.

The authors also observed some unique properties for Rpt1 loop mutants, however, no rational or detailed follow up for that was provided.

Given the lack of substrate-degradation defects for the Rpt1 pore-1 loop mutant, we decided to not further pursue the studies of this variant, despite its increased ATP hydrolysis rate. The complexity of the proteasomal ATPase motor and its regulation for coordinated ATP-hydrolysis, nucleotide exchange, and conformational changes make it extremely difficult to reveal the mechanistic details of this stimulated hydrolysis activity. In light of our cryo-EM structural insights for the s1-state proteasome, where Rpt1's pore-1 loop shows previously missed interactions with Rpt3 that may stabilize the s1 state with low (or no) ATPase activity, we have included in the revised manuscript a brief discussion about the potential roles of Rpt1 and the consequent effects of a Rpt1 YA mutation on the ATPase activity.

Also, in the section on how pore loops affect the conformational dynamics of the proteasome, we now state: *"It is possible that removal of Rpt1's Tyr allows for a less static s1 state that hydrolyzes ATP at a considerable rate without globally switching to non-s1 conformations."*

Overall, the strengths of the paper are the elegant molecular dissection with powerful biophysical assays such as the 26S conformational state with a single molecule FRET-based assay and the substrate translocation with another FRET based assay. These assays have been used in the past and have been further improved upon in this manuscript. These assays are further supported with a cryoEM structure that is of higher resolution as previous structures. This was important to identify the Rpt6 pore loop-Rpt3 Arg finger interactions.

The manuscript is well written and provides strong support for the proposed models and compelling and relevant new insights into the mechanism of proteasome substrate engagement and unfolding. One weakness is that the Rpt1 loop mutant data are largely not explained and show in part phenotypes similar to the Rpt6 loop mutations, suggesting that some of the data can be interpreted differently as Rpt1 loop clearly must be acting differently from the Rpt6 loop.

In contrast to the Rpt6 mutant, the Rpt1 pore-loop mutant showed no degradation defects. The reason for its elevated ATP-hydrolysis activity that does not get further stimulated upon substrate engagement must therefore be distinct from that of the Rpt6 YA mutant. Rpt1's hydrolysis activity is productive, leading to 50% faster degradation of the I27^{V15P} model substrate compared to the wild-type proteasome. While its increased basal ATPase rate may originate from a partial disruption of the s1 state that could allow ATP hydrolysis in some subunits without global conformational transitions to non-s1 states, the elevated ATPase rate and faster substrate degradation by the YA1 mutant in non-s1 states may stem from a reduced bulk in the central channel that allows other subunits to move more quickly. We now mention these hypotheses in the revised manuscript.

Another point of concern is that the general phenomenon that are deduced from these in vitro studies are all based on one model substrate.

Although we agree that it would be insightful to eventually perform similar experiments with a variety of proteins, and ideally endogenous substrates, it was important to conduct our fundamental studies with well-defined model substrates that allow a reliable and reproducible readout. For identifying defects in substrate engagement, unfolding, and translocation, it is critical to know exactly where initiation occurs on a substrate, meaning that there should be no intrinsically flexible or disordered regions other than the unstructured tail. Since the number and location of ubiquitin modifications may influence substrate engagement and degradation rates, we needed a model protein containing a single Lys with a nearby E3 ligase recognition motif that allows efficient ubiquitination in one spot. Also, using the well-characterized titin I27 domain allowed us to generate model substrates with defined thermodynamic stabilities in order to assess how the height of the unfolding barrier affects different proteasome variants in their degradation rates. The titin I27 domain with a cyclin-B-derived tail for engagement is an ideal substrate, and findings regarding different ATPase motor defects should be generally applicable. We do not think that the number and identity of folded domains, or the sequence and location of the unstructured initiation region would fundamentally change the defects observed for the different pore-1 loop mutations.

While the structure of the engagement competent 26S supports the Rpt6 role as gate keeper and thus likely is substrate independent, for some of the other loops that is less clear to me. E.g., the described important role of Rpt4; could there be substrate dependence or other differences where certain loops might play a more dominant role in one case compared to another? Different initiation regions have different efficiencies of degradation, so maybe the length and/or composition of the tail can show more dependence on different loops? So, how can the author be sure the data for Fig. 1D, e.g., is more universal and not specific for this Titin substrate? I think it would be helpful if the authors could explain why this is universal or state the constraints of their study in this regard.

The reviewer is correct that different flexible tails may affect the insertion, engagement, and unfolding, but we don't think that the differential roles of individual Rpt pore loops will change, for 2 reasons. 1) We see defects not just at one particular stage, e.g. initiation, but throughout substrate processing, during which the motor interacts with different segments of the substrate. 2) Our findings about the particular importance of Rpt6 and Rpt4 are nicely supported by structural data and the prominent positioning of these subunits in critical spiral-staircase states. There is just a single engagement-competent s1 state, such that Rpt6 is always at the seam when a substrate lands, it is the first subunit to move in the staircase after substrate insertion, and it is the first subunit to form new contacts with the substrate polypeptide. Similarly, the vast majority of cryo-EM structures for the proteasome (both yeast and human) with stalled substrates or during active unfolding of a stable domain show Rpt4 at the seam or bottom position, which suggests that this subunit has a special role in overcoming an unfolding barrier. Importantly, this is true for various distinct substrates that were used for these structural studies by different groups. Furthermore, previous studies in yeast showed that the Rpt4 pore-1 loop mutant showed the highest accumulation of ubiquitinated proteins in whole-cell lysates, again indicating defects that are independent of the particular substrate identify. We therefore do not think that the use of titin-I27-tail model substrates represents a major limitation of our study or restricts the conclusions that can be drawn from it.

The Rpt1 mutant shows unique and interesting ATPase activity with and without substrate. It is not clear to me how does this mutant fits with the authors' model? Rpt1 mutant has increased ATPase activity for 26S by itself (like Rpt6 loop mutant) without an shift in the population of s1 to non-s1. So, what is going on with Rpt1 mutant and how can authors be sure their interpretation for Rpt6 mutant is correct as there can apparently be different causes of increased ATPase activity? If the higher rate of Rpt1 mutant is for increased activity in the non-S1 state, then combining Rpt1 and Rpt6 mutation should be additive regarding ATPase activity and help distinguish in mechanism between these mutants on 26S by itself?

We currently do not know the exact reason for the increased ATP-hydrolysis and degradation activities of the Rpt1 YA mutant. As mentioned above, it is possible that the YA mutation partially relaxes the s1 state and allows increased ATP hydrolysis without a global conformational shift to non-s1 states that could be detected in our FRET-based

assays. The molecular origin for the increased and productive ATP-hydrolysis in non-s1 states during substrate processing remains unclear as well. Removing the pore-1 loop Tyr from Rpt1 may reduce the bulk in the central channel and allow other subunits to move more quickly, yet have no significant effect on the grip and pulling force of the motor. Our findings from recent cryo-EM studies of the substrate-processing human proteasome would be consistent with this hypothesis (Arkinson *et al.*, bioRxiv 2024 .11.08.622731). In these structures, we observed that Rpt1 and Rpt2 apparently move together from the bottom to the top of the staircase for Rpt2 to re-engage the substrate. During this Rpt2 re-engagement, Rpt5 next to Rpt1 already moves up to the top of the staircase and exchanges ADP for ATP, such that in the subsequent step when Rpt1 re-engages the substrate polypeptide, Rpt5 is nearby and ready to engage. A lack of Rpt1's pore-1 loop Tyr may therefore have no major effects on substrate translocation or unfolding, as neighboring subunits can readily compensate.

Combining the Rpt1 YA and Rpt6 YA mutations, as suggested by this reviewer, will likely not provide further insights, because these mutations affect different stages of degradation and are therefore expected to not be truly additive in their effects. The engagement defects of Rpt6 YA would also apply to the combined mutant and dominate the phenotype. Furthermore, given the interactions of pore loops with each other in the central channel, eliminating more than one Tyr may lead to other global effects that are hard to interpret.

Our new data for mutations at the Rpt6/Rpt3 interface that destabilize the s1 state and increase ATPase activity nicely support our conclusions about Rpt6's pore-1 loop. We are therefore more confident in our interpretations for Rpt6, even if there remain open questions about Rpt1 that may be answered in future studies.

Line 274-275. I like this conclusion by the authors. However, doesn't the Rpt1 mutant suggest this might be too simplistic, as there is higher activity without increased non-s1 state?

In response to reviewers' comments, we mutated E140 and K141 in Rpt6 to disrupt the apparent contacts at the Rpt6/Rpt3 interface that we identified in our new structure of the s1-state proteasome. Indeed, these mutations increased the ATPase activity and the E140A mutation also shifted the conformational equilibrium in our single-molecule assay toward non-s1 states, further supporting our conclusion that the s1 state likely does not hydrolyze ATP. In agreement with this, in our recently solved structure of the resting-state human proteasome, we observed the same Rpt6/Rpt3 contacts as well as pore-1 loops of Rpt1, Rpt2, and Rpt3 held in "parked" positions. The human 26S proteasome has undetectable ATPase activity in the absence of substrate, and cryo-EM particle distributions indicate that it rarely switches to the non-s1 processing states. We now discuss these parallels between the yeast and human proteasomes in the revised manuscript and show the structural overlay in a new Supp. Fig. 4-3.

Mutating the loop of Rpt6 leads to increase non-s1 conformations, which are less suitable for engagement with initiation region. Based on cryoEM the loop of Rpt6 interacts with Rpt3 arginine fingers. The complementary mutations of Rpt3 should thus also change the equilibrium from more s1 to more non-s1, which could be a nice confirmation of their data,

assuming the mutated arginine fingers don't interfere too much with ATP binding and ATPase ring formation. Did the authors attempt or consider this mutation?

The reviewer is correct that mutations of the Arg fingers would compromise ATPase activity and are therefore not suited to address this question. Instead, we introduced the E140A and K141A mutations in Rpt6, and indeed observed the expected effects: both mutants show an increase in ATP hydrolysis activity, and our single-molecule conformational change measurements for the E140A mutant showed a corresponding shift from the s1 to non-s1 states (new Supp. Fig. 4-4). As mentioned above, we also present in the revised manuscript a direct structural comparison with the s1 state of the human proteasome, which shows the same Rpt6/Rpt3 contacts and interactions with the pore-1 loops of Rpt1, Rpt2, and Rpt3 that likely help stabilizing a static, ATPase deficient s1 state.

The supplementary Fig 7-2 C provides for nice complementary data to the other type of assays. However, looking at the figure the impact seems rather modest compared to the described impact of the mutations from the other assays. Wouldn't it be expected that more of the substrate would be degraded in the wild type versus Rpt6 and Rpt4 mutants, this is not apparent to me from this assay.

The reviewer is correct about these SDS-PAGE results being not as clear as expected based on our other assays. We were also disappointed to not see a stronger accumulation of partially degraded product. In part this may be due to a lack of tail clipping when the motor fails to unfold titin I27, leading to the release of deubiquitinated full-length substrate, rather than a truncated product (see our response to the next point below). Furthermore, it is possible that a small amount of free 20S core particle from dissociated 26S proteasomes truncates the tail of released substrates, generating non-uniform products that spread over a wider range of the gel.

Regarding the extend of substrate degradation for wild-type versus Rpt6 and Rpt4, this is always hard to assess by SDS-PAGE, since there is a broad distribution of ubiquitinated substrate species, and wild-type titin I27 is turned over very slowly. The single-turnover bulk degradation measurements (Supp. Fig. 6-2) and the single-molecule analyses of WT titin I27 degradation (Fig. 6, Fig. 7, Supp. Fig. 6-3) are therefore more reliable in determining differences in degradation kinetics for WT proteasome versus its YA mutants, and WT proteasomes show faster turnover there. It is indeed possible that the less frequent substrate release during or following an unfolding attempt hold up WT proteasomes, such that YA4 and YA6 mutants may more quickly move on to the next substrate and thereby generate more truncated as well as deubiquitinated substrate species. This is indeed what the gel in Supp. Fig. 7-2 C seems to show: YA4 and YA6 appear to turn over more ubiquitinated substrate, and the bands for deubiquitinated and deubiquitinated + truncated titin are more intense than for WT proteasomes. We briefly discuss this in the revised manuscript and thank the reviewer for pointing this out.

Also, would all the released substrates be truncated or could some be full length and only be deubiquitinated?

The reviewer is absolutely correct. Only if the C-terminal unstructured tail of 59 is reasonably extended within the degradation chamber of the 20S CP by the time the motor reaches the titin I27 folded domain, one would expect clipping prior to release. Failure to unfold may therefore also lead to deubiquitination and release of full-length substrate. Although this is harder to quantify, as there is some unmodified substrate in our initial sample due to incomplete ubiquitination, the gel in Supp. Fig. 7.2 C indeed appears to show a more intense unmodified titin band for YA4 and YA6 mutants compared to the WT proteasome, which we now briefly discuss at the end of the results section.

Reviewer #2 (Remarks to the Author):

This manuscript by López-Alfonzo, Erika M., et al., presents a comprehensive study using single-molecule FRET and cryo-EM techniques to elucidate the role of pore loops in the Rpt6 and Rpt4 ATPase subunits of the 26S proteasome. The authors provide valuable insights into how these structural elements affect the protein degradation process, demonstrating that the Rpt6 pore loop is crucial for substrate handling and initiation of degradation, whereas the Rpt4 loop aids in substrate unraveling and sustains proteasome efficiency. Although the experiments are well-executed and the findings presented are intriguing, I have the following major concerns:

1. The study explores the Rpt6 pore-1 loop's role via the YA6 mutation, positing that it affects substrate capture. Could you detail whether this mutation alters the interaction between Rpt3 and Rpt6?

The reviewer is correct, the YA6 mutation indeed weakens the s1-conformation-specific interaction between Rpt6 and Rpt3, leading to a shift in the conformational equilibrium toward non-s1 states. Our new mutations E140A and K141A in Rpt6 confirm this finding by also weakening the interaction with Rpt3, leading to an increase in ATPase activity and, as tested for the E140A mutant at the single-molecule level, to a shift in the conformational equilibrium. Depending on the mutation, destabilizing this interface may also have effects on proteasome assembly, as alternative Rpt3 mutants, D304A and D325A, showed significantly reduced yields and were therefore not further pursued for the revision of this manuscript.

Furthermore, how does the helical conformation of the Rpt6 pore-1 loop interact with the arginine finger of Rpt3 to facilitate substrate capture and the conformational state transition?

The helical conformation of Rpt6's pore-1 region and its interaction with the neighboring Rpt3, including the pore-2 loop and the second region of homology with its Arg fingers, seems to stabilize the s1 state and hold the proteasome in a resting state until substrate arrival. Based on our observations for both the yeast and the human 26S proteasomes, we assume that in the s1 state the spiral staircase is static and there is very little or no ATP hydrolysis. In the revised manuscript we now present that yeast and human proteasomes share the same interactions that "park" the ATPase motor. Those interactions involve not just the Rpt6/Rpt3 interface, but there also appear to be stabilizing

interactions (mostly hydrogen bonds) between the Rpt2's pore-1 loop Tyr and Rpt3, between the Rpt1 pore-1 loop Tyr and Rpt3, as well as between the Rpt3 pore-1 loop Tyr and the N-domain. It remains to be determined by what mechanisms substrate insertion into the central channel breaks those interactions, leading to engagement and a transition of the motor from a static s1 to dynamic non-s1 states with actively hydrolyzing subunits. Based on our single-molecule conformational change assay, the yeast proteasome spontaneously samples non-s1 states, which may allow Rpt6 and other subunits to grab a substrate that inserted itself in the channel. Although there are no similar single-molecule studies for the human proteasome, particle distributions for cryo-EM data sets and undetectably low ATPase activity in the absence of substrate indicate that the human proteasome does not frequently switch to non-s1 states, and a specific mechanism may trigger this transition upon substrate insertion.

2. The manuscript references data on ATP γ S in Figure 2's legend that appears absent from the figure itself. Could you include a supplementary explanation or correct the figure to reflect this data?

The reviewer is right about the incorrect legend for Fig. 2 C-E. This has been corrected. The ATP γ S data are shown in the supplement (Supp. Fig. 3-4).

3. You mention kinetic parameters from BNP-FRET analysis for transition rates; however, the methodology to derive these figures is unclear. The visual distinction in Figure 2D between states is not evident. Could you elaborate on the criteria used for determining these rates?

We thank the reviewer for pointing this out. We have now clarified the procedure for generating the figures in the text. Briefly, BNP-FRET works in the Bayesian paradigm where it generates a collection of Monte Carlo samples to estimate probability distributions for each parameter of interest, including rates and trajectories. The heatmap in Fig. 2D has contributions from the whole ensemble of FRET traces and a wide distribution reflects the kinetic heterogeneity among traces. This, in turn, makes it challenging to identify states at the ensemble level.

4. In Figure 2E, the fitting of k_{fast} and k_{slow} for the escape rate of the low-FRET s1 state suggests two kinetic processes. If these rates are not indicative of separate processes, please clarify their relevance.

The fast transition rates are dominant, which is why we largely disregard the slow rates, whose molecular origin remains unknown. We now mention this explicitly in the revised text.

5. The authors have excluded static fluorescence traces from the population analysis. It would be beneficial for the study if the reasons for these traces exhibiting static characteristics were discussed. Specifically, it should be clarified whether these static traces originate from denatured samples, which could potentially exhibit different fluorescence behaviors. If the static nature is not due to denaturation, an explanation

should be provided regarding why these samples exhibit such characteristics. This discussion could help validate the exclusion of these traces and ensure that the analysis accurately reflects the behavior of functional protein complexes.

We appreciate this reviewer's critical assessment of this issue, which represents a common challenge in single-molecule analyses. We disregarded static traces based on the more consistent behavior we observed for dynamically switching proteasomes, combined with results from bulk measurements and with cryo-EM data showing certain distributions of s1 and non-s1 states for substrate-free yeast proteasomes. It seems not surprising that some of the immobilized molecules are "sick" and not behaving as expect. As suggested by the reviewer, denaturation, partial unfolding, or sticking may play a role. Based on the fluorescence signal we can rule out oligomerization or aggregation of several labeled particles, but otherwise it is impossible to say why some proteasomes appear stuck in a certain state. We now explain in the text why we focus on the dynamically switching molecules and disregard the static ones.

6. The Rpt1 mutant YA1 shows enhanced ATPase and degradation activities. Could further insights into the mechanistic implications of this observation be provided, particularly regarding the pore loop's role as a potential rate-limiting factor?

We do not know what is rate limiting for the motor movements during substrate unfolding and translocation, but it is possible that the YA1 mutation removes bulk from the central channel in non-s1 states and makes the motor overall move more quickly. Importantly, the increased ATPase activity also leads to increased degradation velocity for all the substrate variants, indicating that this mutant can hydrolyze and move faster, with no apparent negative consequences on substrate processing, which makes this distinct from Rpt6. We now discuss in the revised manuscript the potential origins for the increased ATPase activity of the YA1 mutant in the s1 and non-s1 states.

7. The role of the YA6 mutation in facilitating transitions to non-s1 states lacks direct evidence linking substrate interaction directly with Rpt6.

The effect of the YA6 mutation may be direct, through Rpt6's pore-1 loop contacting substrate, or indirect, through a compromised interface with Rpt3. It is hard to assess what contributes what. From structures during active processing it is clear that Rpt6 makes direct physical contact with the substrate, but when and how Rpt6's pore-1 loop is released from its "parked" position against Rpt3 remains unclear. The yeast proteasome spontaneously switches from s1 to non-s1 states more than twice per second, such that Rpt6's pore-1 loop may grab an inserted substrate for engagement. The resting (s1) state of the human proteasome is much more static and rarely switching to non-s1 states, such that substrate insertion into the central channel may trigger a conformational transition and release of Rpt6's pore-1 loop from Rpt3, yet elucidating the underlying mechanism is beyond the scope of this paper.

Structural data in Figure 4 imply dependency on Rpt3. Could the mutation's effects be due to destabilizing Rpt3/Rpt6 interactions?

Yes, the reviewer is correct about this destabilization of the s1-specific Rpt6/Rpt3 interface by the YA6 mutation. We were able to further support this model by characterizing two other interface mutations, Rpt6 E140A and K141A. Both show increased ATPase activity, and the E140A mutation tested in our single-molecule assay leads to a shift of the conformational equilibrium towards non-s1 states, similar to the YA6 mutation. We also attempted to mutate the Rpt3 side of this interface by replacing D304 or D325 with Ala, yet these mutations led to significantly reduced yields of the base subcomplex (possibly due to assembly defects), and were not further pursued for this revision.

Direct structural evidence of substrate engagement with Rpt6 would strengthen this claim.

Although there is structural evidence for Rpt6 physically interacting with substrate at later stages of processing, it seems impossible to catch this subunit in the process of initial engagement.

8. The observation of increased substrate releases due to mutations in the Rpt4 pore-1 loop suggests a gripping defect. Could an additional mechanism explain this loop's role in substrate processing?

Issues with stably gripping the substrate during mechanical unfolding seem to best explain the phenotypes we observed for the YA4 mutation. Our single-molecule measurements provide unprecedented insights that go far beyond of previous bulk biochemistry or *in vivo* data for the contributions of individual ATPase subunits to proteasomal substrate degradation, but of course we cannot rule out alternative or additional mechanisms.

9. You propose a "burst-like" mechanism contrasting with the traditional hand-over-hand model. How does this align with the asymmetric arrangement of Rpt subunits and the structure of individual subunits? More data would bolster this novel hypothesis.

The hypothesis of the "burst-like" mechanism is based on our functional data, combined with strongly asymmetric distributions of proteasome particles in different spiral staircase states that were observed in various cryo-EM data sets. Our recent structural studies of the human proteasome during active substrate degradation strongly support this model, as they indicate that movement of Rpt4 from the bottom to the top of the spiral staircase is the slow step during the ATPase cycle of the motor hexamer and is linked to ATP hydrolysis in two Rpt subunit, Rpt2 and Rpt1. This movement of Rpt4 may therefore represent the power stroke of the proteasomal motor for substrate unfolding.

We cite this work now in the revised manuscript.

10. The use of singly fluorescently labeled substrates could potentially influence degradation rates. Have comparisons been made between rates of ATP hydrolysis and protein degradation in fluorescently labeled versus non-labeled substrates?

The reviewer brings up a very valid concern. For previous bulk and single-molecule studies (Bard et al., 2019; Jonsson et al., 2022) we tried to compare labeled and unlabeled substrates. The rate-limiting step of substrate degradation was not affected, but translocation through the proteasome central channel may have been somewhat slowed by pulling through a bulky fluorescent dye. However, direct comparisons are difficult, because unlabeled substrates cannot be analyzed regarding the individual phases of their degradation. Importantly, we observed robust and rapid turnover of the fluorescently labeled substrates, and by using identical substrate batches for the characterization of different proteasome mutants we assured that the measured effects are reporting true functional differences.

11. To substantiate the claim that mutants YA4 and YA6 destabilize the non-S1 state with the substrate (Figure 5J), could you provide dwell-time distributions for high-FRET states across other mutants?

We now include in the new Fig.5J the High-FRET dwell time distributions for all YA mutants, showing that only the YA4 and YA6 mutations lead to significantly shorter times continuously spent in non-s1 states during substrate processing.

12. In Figure 6, the definition of FRET states shows significant fluctuations, especially in the unfolded state (Figure 6D). Could you provide a detailed methodology for how these FRET states were determined and assigned?

The pre-unfolding FRET state was assigned based on comparing WT and YA mutants and comparing WT titin and the titin V15P mutant, with the latter not showing much of the pre-unfolding intermediate FRET phase. Also, Supp. Fig. 6-5 shows slips out of the pre-unfolding intermediate FRET to a higher FRET state and back. Overall, we think that there is reasonably solid evidence for the lower FRET state being a pre-unfolding dwell. We now mention in the revised manuscript that this assignment as a pre-unfolding dwell was made based on the distinct presence for the thermodynamically much more stable wild-type titin, while being much shorter or absent for the destabilized titin-V15P mutant.

Reviewer #3 (Remarks to the Author):

Using FRET-based single-molecule analysis, López-Alfonzo et al. examined six Pore-1 loop mutants, where the conserved Tyr residues in the Pore-1 loop were substituted with Ala, revealing that the Pore-1 loops of Rpt6 and Rpt4 play unique and distinct roles compared to other subunits. They developed an algorithm to improve the analysis of the previously established smFRET method, enabling unbiased assessment of structural fluctuations. Structural analysis using Cryo-EM demonstrated that the Rpt6 Pore-1 loop, whose mutation resulted in defects in both ATPase activity and substrate degradation assays, forms an α -helix in the s1 state, accompanied by a unique Arg finger orientation. Overall, the experiments were well-designed, with the structural dynamics derived from their FRET analysis aligning with those observed in Cryo-EM. The established smFRET analysis provides significant and unique insights into the mechanical function of the

proteasome. While a few minor questions remain, the study is well-suited for publication in Nature Communications.

We thank this reviewer for the overall very positive evaluation of our study.

1. YA6 increases the non-s1 conformation in the presence of ATP, while no significant changes are observed in YA1 and YA4. The escape rate, representing the rate of conformational switching, is notably faster than the ATPase hydrolysis rate, implying that the conformational switch is independent of hydrolysis.

What, then, triggers the conformational switch? Additionally, how does hydrolysis influence this process?

The reviewer is correct about the conformational switching occurring potentially spontaneously. In fact, we now postulate that the resting s1 state does not hydrolyze ATP at all, and the basal ATPase rate may originate solely from the spontaneous switching to hydrolysis-active non-s1 states. This assumption is not only consistent with the single, apparently static spiral staircase orientation observed in structures of the s1-state proteasome, but also supported by the undetectably low ATPase activity of the human proteasome in the absence of substrate, which does not show significant spontaneous switching to processing non-s1 states in cryo-EM particle distributions. We now reference those recent studies of the human proteasome.

The molecular reason for the conformational switching remains unknown. It could just be a stochastic thermal crossing of the energy barrier between s1 and non-s1 states in this conformational equilibrium, or switching could be triggered by an ATP hydrolysis event in the s1 state. Regarding the latter, it is interesting that ATP γ S induces and stabilizes the non-s1 states, indicating that nucleotide identity in some of the ATPase subunits strongly affects the global conformational equilibrium.

Substrate engagement appears to drive the conformational switch or at least stabilize the non-s1 states, likely by bridging 4-5 ATPase subunits through pore-1 loop contacts and coordinating or synchronizing their conformational changes and ATP hydrolysis. The observation that the human proteasome quickly engages substrate despite the absence of spontaneous switching to non-s1 states suggests that there are trigger mechanisms through substrate interactions in the s1 state, and engagement is independent of ATP hydrolysis or spontaneous switching.

2. In Cryo-EM analysis, it is important to show the density map to validate whether the side-chain structures are accurately modeled.

We apologize for not including such density maps with docked atomic models in the original paper. The new Supp. Fig. 4-2 now includes those examples for density maps.

3. Among the structures of the yeast 26S proteasome, s2 and s5 exhibit similar configurations to the s1 state, with Rpt6 positioned as the seam subunit. Do these structures share loops that form a helix with Tyr222 oriented inward? Additionally, can this helix formation of the pore-1 loop be observed when other subunits occupy the seam position? A comparison would be interesting.

Only the s1 state shows this Rpt6-3 interaction with a helical conformation for Rpt6's pore-1 loop. None of the other interfaces, neither in s1 nor in non-s1 states, including seam subunits, show similar conformations and interactions. We are trying to highlight this in the Supp. Fig. 4-3, depicting an example non-seam interface (Rpt2/Rpt6) in the s1 state and a seam as well as non-seam interface (Rpt4/Rpt5 and Rpt6/Rpt3) in a non-s1 state.

4. smFRET data showed that the YA4 mutant significantly increases the s1 conformation in the presence of substrate, presumably due to substrate slip. The reviewer questions whether a single mis-capture could indeed lead to entire substrate release. What is the probability that a mis-capture of the substrate causes its release? Would it be feasible to analyze the frequency of substrate release to investigate this further?

The reviewer raises an interesting question: Is a slip in front of the unfolding barrier sufficient for allowing the entire tail to slide out of the central channel and make a substrate escape degradation. In Fig. 7B we show the substrate release frequency after unsuccessful unfolding attempts and in Fig. 7C we show the time spent trying to unfold (successfully and unsuccessfully). Based on these results, a single slip does not seem sufficient for release. We see a lot of s1 transitions for YA4 during substrate processing, meaning it can slip and maintain the substrate bound. Based on Fig. 5J, YA4 slips and switches to s1 with a time constant of 1.8 s, yet based on Fig. 7C it spends on average 15 s trying to unfold a substrate before giving up and releasing. A single slip consequently does not lead to release, and it takes several slips until a substrate escapes. We now discuss this at the end of the results section and thank the reviewer for bringing this up.

5. A more detailed discussion on the dysfunction of YA4 would be valuable, as the data plausibly suggest that Rpt4 acts as a barrier to switching motion. In Cryo-EM analyses, s1-like and s4-like conformations are frequently observed in the absence and presence of substrate, respectively. This may result from the instability of other seam configurations. However, it remains unclear how the Rpt4 loop actively facilitates the transition from the bottom position to the seam.

Our recent structural studies of the substrate-processing human proteasome agree with the findings here and in previous cryo-EM analyses that Rpt4 frequently resides at the bottom or the seam of the staircase when the proteasome is dealing with an unfolding or translocation barrier (Arkinson *et al.*, bioRxiv 2024 .11.08.622731). We observed reproducibly more than 50% of proteasome particles with Rpt4 at the bottom or the seam, which we now include in the revised manuscript.

Together with the detected nucleotide occupancies and transitions between states, we proposed a burst-like mechanism in which some subunits move in rapid succession or even simultaneously, whereas other subunits, i.e. Rpt4, progress more slowly. Although we do not have concrete proof, it is conceivable that staircase conformations with Rpt4 at the seam or the bottom of the staircase represent pre-power stroke states, in agreement with our findings here that Rpt4's pore-1 loop plays a particularly important role for substrate unfolding. However, the molecular origins for this asymmetry in burst-like subunit movements and their contributions to unfolding remain unknown.

Reviewer #4 (Remarks to the Author):

Kinetics analysis for single-molecule FRET traces

> To analyze the FRET transition kinetics, authors used the Hidden Markov Model, a well-established model for single-molecule analysis. They used "ebFRET" software for this, and transition density plots show the transition between two dominant FRET states for the proteasome construct. They also used Bayesian nonparametric FRET-analysis algorithm based BNP-FRET which does not assume the number of states. The authors were to better explain how transition rates were obtained and if the BNP-FRET result was different from the ebFRET result.

We thank the reviewer for this comment. In response to another reviewer's comment, we have elaborated upon the procedure for generating rates. Briefly, ebFRET and BNP-FRET do not contradict each other, rather they complement each other. BNP-FRET generates Monte Carlo samples for all parameters of interest from all FRET traces, which are then combined to produce the probability distributions over rates, trajectories, and FRET efficiencies. At the ensemble level, trace-to-trace kinetic heterogeneity causes the distribution over rates to be wide, making it difficult to identify states. At the individual trace level, ebFRET estimates lie within the high probability regions of distributions estimated by BNP-FRET.

Comments on the transition rate analysis

They were to clearly specify the used fitting model (single or double-exponential) and overlay the fitting curve with the dwell time data in Supp. Fig.2-2 or elsewhere to show if their model actually fits. It seems that they used double exponential fitting to get the "fast" and "slow" rates.

We specify in the figure legends (e.g. Fig. 2E, Fig. 3G,H, and new Supp. Fig. 3-3) that transition rates were determined by double exponential fitting of the dwell time distributions. The new Supp. Fig. 3-3 show the dwell time distributions for the s1 and non-s1 states of WT proteasomes as well as the YA4 and YA6 mutants, and their fits to double exponentials to determine the fast and slow phases of transition rates (shown in Fig. 3G,H).

Because we expect the single-exponential distribution (with a tail...) in the simple two-state system (assuming one dominant rate-limiting step) and double exponential fitting usually fits many types of data, we need some justification for the double exponential fitting. Is it due to the limited temporal resolution or another technical limitation? Or, is a two-step reaction actually expected here?

We assume that these deviations from ideal behavior are due to the complexity of the multifaceted system, including a not completely uniform substrate pool, slight differences in ubiquitination, or compromised activity of immobilized proteasomes. Some molecules

may not be as active. We do not assume a 2-step reaction, although it cannot be completely ruled out.

Comments on each line

220: with rates of 0.34 - 4.22 s⁻¹ ... 0.66 - 7.22 s⁻¹

> What does the range mean? Confidence interval? The range is not very clear or informative for a data set with an intrinsic wide distribution...

We thank the reviewer for this comment. We have now clarified in the text that the range in the ensemble heatmap indicates kinetic heterogeneity rather than uncertainty in rates for a given trace.

222: According to the ebFRET analysis (Fig. 2E)

> Fig. 2E legend says the data is Bivariate probability distribution (not ebFRET). Which one is correct?

Thank you for pointing out this mistake. The legends for Fig. 2 C-E were wrong and have been corrected.

> Describe what three dots represent, here. t1/t2 of double exponential fitting of each data set? How can we get that from Fig. 2D?

The 3 dots represent the values determined in 3 technical replicates with separate ebFRET analyses, which is now much clearer with the correct figure legend.

Fig. 2D/2E: The data should be ATP vs ATP_rS, but there is only one graph for each subplot.

Again, this was our mistake with an incorrect figure legend. The data for ATP_γS are shown in the supplement (Supp. Fig. 3-4).

225: The rates determined by ebFRET and BNP-FRET show overall similar trends and the observed differences ... using camera calibration frames to avoid overfitting.

> I believe the author should find a better analysis (may depend on the parameter we want to see) and provide a fair justification for using the method over the other one. If the difference is from the various noises, we cannot trust the data for the analysis.

We thank the reviewer for this comment. In response to an earlier comment and this one, we have added more clarification points in the text. We use ebFRET and BNP-FRET as complementary tools that confirm each other's conclusions, although one is more general and newer than the other. Furthermore, we now emphasize further in the text that all microscopy data are corrupted by noise, including photon shot noise, camera noise, and spectral crosstalk. By calibrating our camera and other sources of noise, BNP-FRET provides accurate uncertainty estimates for rates to complement deterministic estimates by ebFRET.

In my opinion, because BNP-FRET does not assume the number of states, the result can be less clear. If we can confirm that there are only two states based on the traces, the ebFRET result can be more clear. BNP-FRET can be used to confirm if there are 3rd or more states.

We appreciate this observation by the reviewer. We have now clarified further in the text that, although we observe two FRET efficiencies at the individual trace level, it may not immediately imply the existence of only two states. We do observe trace-to-trace kinetic-heterogeneity indicating the possibility of a rugged energy landscape for conformational changes. In other words, more than two conformations likely exist, but are degenerate in FRET efficiency. Furthermore, existence of rarely visited intermediate FRET efficiency state would manifest itself as higher uncertainty/wider distribution in BNP-FRET estimated rate vs FRET efficiency heatmaps, regardless of whether the intermediate state shows up in an individual HMM trajectory.

238: This was confirmed in the BNP-FRET analysis by a redistribution of the s1-to-non-s1 escape towards higher rates that extend beyond 6 s⁻¹ (Fig. 3E.. > IT seems that both states got longer escape times in Fig. 3E, which means that the transition got slower which does not necessarily mean that the high FRET state is more populated. We'd better check the numbers or the bar graph. The single-molecule distribution of dwell time (escape time or rate) is informative. It is hard to follow the distribution in the 2D density plot for me.

The reviewer is correct that there is an increase in the distribution of both escape rates for the YA4 mutant (Fig. 3E), meaning that some population appears to switch more rapidly. However, the mean values for switching rates determined by ebFRET, shown in the bar graph in Fig. 3G + H, indicate that YA4 changes conformations with similar rates as the wild-type proteasome.

Fig3

> Labeling non-S1 or S1 on each FRET peak would help.

We thank the reviewer for this suggestion. This has been done.

Fig 3G/H

> a color legend for WT/YA6/YA4 would help.

This makes sense. Done.

243 The more frequent sampling of processing-competent non-s1 states by the YA6 mutant in the absence of substrate is also consistent with its higher basal ATPase rate. > As mentioned at 251, YA1 is not like this. It is not convincing... we may need a better explanation or insight on the structure.

We agree with the reviewer that it is necessary to better explain the potential scenarios that could lead to this differential behavior of the YA1 and YA6 mutants. In the revised manuscript we discuss in more detail that the removal of Rpt1's pore-1 loop tyrosine may increase the flexibility of the s1 state and allow ATP hydrolysis in some subunits without a global conformational switch to non-s1 states. The faster substrate processing by the YA1 mutant may be explained by a reduction of crowding in the central channel that allows other subunits to move more rapidly, yet does not affect the grip or pulling force on a substrate. The increase in productive ATP hydrolysis and degradation velocity when in non-s1 states will be hard to study structurally. Solving a cryo-EM structure of the YA1 mutant in the s1 state may provide some insights into the increased basal ATPase rate (possibly due to a partial disruption of the network of hydrogen bonds formed by the pore-1 loop Tyr of Rpt1, Rpt2, Rpt6, and Rpt3 with Rpt3), but such a structure is unlikely to reveal the reasons for increased substrate processing in non-s1 states and is therefore not really worth pursuing. We hope that this reviewer agrees that the present study should primarily focus on the Rpt subunits with the strongest contributions to proteasomal substrate processing, i.e. Rpt6 and Rpt4. We think that through the combination of *in vitro* biochemistry, single-molecule measurements, and cryo-EM structure determination, we could provide a reasonable model for AAA+ motor mechanisms that is well supported by the experimental data, even though some questions, for instance about the stimulated ATPase activity of the YA1 mutant, remain unanswered.

285: We previously found that upon substrate engagement ... high-FRET non-s1 states that are maintained until substrate unfolding and threading through the ATPase
> Based on this, we will assume that high-FRET state dwell is the sum of intrinsic high FRET, substrate folding time (after binding), and threading time.

Theoretically yes. But it is tricky, because numbers come from conclusions drawn using 2 different assays, each with their shortcomings (peptide release from 20S holds things up in the processing assay, whereas spontaneous switching may make processing look shorter in the conformational switching assay).

288: With the new instrumental setup

> The author mentioned this multiple times. Need to specify what was improved technically (eg. temporal resolution?) and what information authors additionally obtained. Or remove it.

We removed it.

289: high-FRET phases, i.e. substrate-processing dwells

> unfolding and threading time (285)

Following the reviewer's suggestion this has been changed to be more specific: "*high-FRET phases, i.e. substrate-processing dwells that include unfolding and complete threading*".

299: we were previously unable to use ebFRET for reliably measuring the ...

> I do not get the difference between ebFRET and BNP here. Both measure the dwell time of each state. It seems that the bivariate probability distribution of BNP-FRET shows the distribution of the "rate" (the inverse of "dwell time") of each event. If so, there should be no significant difference between the two methods in principle. If not, bivariate probability distribution should be better explained.

We have clarified these points in response to previous comments.

> It is not easy to compare bivariate probability distributions. In this figure. The sum of all pixels should be 1, but Fig 5I has much more bright pixels compared to 5G. It seems that they are not normalized equally or the color scale is not showing the proper region.

We thank the reviewer for bringing up this observation. We double checked, and the plots indeed represent the correct probability densities, shown in log space in order to make low-density regions more visible. Due to the log space representation, normalization is not trivial and intensities appear not additive.

306: higher rates of 0.38 - 8.72 s⁻¹, compared to 0.55 - 6.61 s⁻¹ for WT
> not a good way to represent the data

We have addressed this in response to this and another reviewer's comment.

309: correspondingly wider range of rates
> "wider" what does it mean? Is the rate more heterogeneous or higher?

The rates are overall higher, and we changed the text accordingly.

315: To more specifically analyze how
> need to specifically explain the benefit of the manual scoring...
> Was the manual scoring result different from the ebFRET or BNP? With the slower rates, both software may find the state better, too.

In particular the analyses by ebFRET benefitted from the lower substrate concentration that allowed a better distinction between degradation events and phases of spontaneous conformational switching without substrate. In the revised text we added the following sentence: *"This lower substrate concentration reduced the frequency of degradation events and made it easier to distinguish between phases of degradation and idling without a bound substrate."*

320: shorter high-FRET phases, with $\tau_{YA4} = 3.2 \pm 0.3$ s and $\tau_{YA6} = 3.1 \pm 0.3$ s
> Specify the representative numbers. Are they mean?

Yes, these values are the mean \pm standard error of the mean, which we now specify in the revised text and it is also mentioned in suppl. table 8.

> I suggest the author show the distribution of dwell time (frequency vs dwell-time) as

many conventional single-molecule data and discuss the distribution more. It seems that WT has a more Gaussian distribution which implies that the high FRET dwell time is the result of multiple reactions, or threading. YA6 and YA4 are not. It seems that the result is a mixture of failing (single or double exponential) and some threading.

This is a very good suggestion. We now included a new Supp. figure 5-4 that shows the cumulative frequencies of high-FRET dwell times for WT and all YA-mutant proteasomes and reveals a clear deviation for YA4 and YA6, whose majority of high-FRET phases during degradation last less than 2 s.

327: Rpt4's pore-1 loop affects the stable maintenance
> Specify how it was affected

This sentence was rephrased to now read: "...mutation of either Rpt6's or Rpt4's pore-1 loop compromise the stable maintenance of the processing-competent, non-s1 states during substrate unfolding and cause more frequent returns to the s1 state, potentially due to slippage."

Fig. 5C-E
> labeling (WT, YA6, YA4) would be nice.

Done.

353: slower substrate insertion into the YA6 mutant can be explained by its shift of the conformational equilibrium
> YA4 showed the slower insertion, but its shift of the conformational equilibrium is the opposite.

We changed the text to now say that the slower substrate insertion into the YA6 mutant can "to some extent" be explained by the shift of the conformational equilibrium, and we further discuss that the YA-mutation-induced dislodging of Rpt6's pore-1 loop from Rpt3 may sterically interfere with the diffusive substrate insertion into the central channel. The slower substrate insertion into the YA4 mutant may have other reasons, for instance a less stable commitment to substrate degradation at a slightly later stage, for instance due to failed power strokes that drive substrate translocation after engagement, as discussed at the end of the results section on capture success.

360: suggesting that substrates either do not fully enter the ATPase ring or fail to stay stably inserted
> If so, oPA does not work for mutants and we cannot assume the single-turnover scenario. Is it still fair to accept the measured time constant?

We can still consider this as a single-turnover scenario regarding successful substrate engagement, even if this engagement takes several attempts and not all proteasomes end up with a stably engaged substrate. oPA does still work, as it prevents the degradation process to proceed beyond engagement. It is therefore fair to accept the

measured time constant, for the same reasons why we can determine degradation time constants for the proteasome, despite substrate capture by the motor being successful in just ~ 50% or less of interaction events and requiring re-binding of the substrate.

1072: Traces were filtered for visualization using Matlab R2019b.
> Specify the filtering criteria instead of the software.

We used the MATLAB R2019b moving average filter, which is MATLAB's default smooth function with a heuristically determined averaging window of five frames (equivalent to 250 ms). We now provided additional information in the legend for Fig. 6.

More minor points

184: Phe2
> What is Phe2?

Phe2 stands for phenylalanine 2. Depending on the context, we use single-letter code (for mutations), three-letter code (specifying certain residues), and the full name for amino acids, but we will adhere to the journal style if a certain code is preferred.

592: 5,50-dithiobis-2-nitrobenzoic acid
> 5,5-thiobis?

Thank you for catching it. This has been corrected.

627: 300 uM LD555 (Lumidyne Technologies)
> DBCO? Cat No. Can be added to all reagents. Specify dyes and their functional group as possible.

We now explain that DBCO-LD555 and DBCO-LD655 are dibenzocyclooctyne-conjugated sulfo-Cy3 and sulfo-Cy5 derivatives, respectively, from Lumidyne Technologies, with the DBCO modification being custom synthesis. Due to the custom synthesis there is no Cat.No.

680: Excess dye was quenched
> The dye was not quenched here. The reaction was.

This was corrected.

742: built on an ultrastable single molecule stage
> There are many words for optical tables. Single-molecule stage sounds a bit weird.

We changed this to "ultrastable optical table".

746: at 5mW and 2mW power
2mW laser (from the laser head) is usually not good enough for single-molecule imaging in general. Better specify the information.

Thank you for pointing this out. This information has been corrected. We used 40% power for the 532 nm laser and 25% power for the 640 nm laser. The output power at the laser had / fiber optics is ~ 20 mW, meaning that at the sample stage we had ~ 8 mW for 532 nm and 5 mW for 640 nm.

793 but with a higher EM gain of 40
> The 40 cannot be a real gain of iXon for smF...

We double checked, and this was indeed the gain value we consistently used for the EM Gain, together with a preamp setting of 1 and a readout rate of 17MHz

Supp. Fig.2-1:
> The camera model number is wrong.

Thank you for pointing this out. The number has been corrected to iXon 897.

Supp. Fig.2-2: "hmm-mcmc"
> "hmm-mcmc" was not mentioned before. It should be a BNP-FRET-related one. Please use consistent terminology.

We corrected this mistake. Thank you.

"DTNB was added to the purified base after concentrating"
Is it to minimize the non-specific interaction of DBCO? Does blocking thiol affect the protein function?

The reviewer is correct that the reaction of exposed Cys with DTNB protects them from the undesired modification with DBCO. Blocking thiols this way may have effects on protein function, and we therefore reverse this modification by subsequent incubation with DTT, which restores free thiols. We now explain this in the revised methods section.

Q. The construct shows the transition between two states (S1 or non-S1). Do substrates bind to both states?

Both states bind substrate through the interaction of substrate-attached ubiquitin chains with proteasomal ubiquitin receptors, but only the s1 state allows a substrate to enter the central channel and be engaged by the ATPase motor.

Non-S1 states process a substrate after successful engagement, but do not allow efficient substrate insertion due to steric occlusion of the pore by Rpn11. We added a sentence in the introduction to further clarify this.

Response to reviewers' comments:

We thank all 4 reviewers again for their constructive criticism that helped us to significantly improve our manuscript for publication in *Nature Communications*.

Our response to reviewer #4's concern about experimental repeats is below.

Reviewer #1 (Remarks to the Author):

Authors have addressed all my concerns and questions.

Reviewer #2 (Remarks to the Author):

The authors have addressed all of my concerns. I have no further suggestions and recommend the manuscript for publication.

Reviewer #3 (Remarks to the Author):

By responding thoughtfully to my comments and that of the other reviewers, the authors have greatly strengthened and improved the manuscript. This paper is exceptional in both its biophysical approaches and its analysis of proteasome functions, and it is fully deserving of acceptance in Nature Communications.

Reviewer #4 (Remarks to the Author):

The authors' response and the new revision are acceptable.

One remaining minor concern is related to Fig. 2E.

The statement, "Three technical replicates with separate ebFRET analyses, contributing to N = 289 particles," could be interpreted as indicating that the main figure is based on a single experiment whose dataset was divided and analyzed three times, rather than on three independent biological experiments. If this is the case, the data may not be sufficiently robust for inclusion as a main figure. If the authors indeed performed three independent experiments, it would be clearer and more appropriate to state this explicitly.

Response:

We apologize for not presenting this more clearly. The data presented in Fig. 2E were not just from a single experiments or from technical repeats, but from biological repeats, using reconstituted proteasome subcomplexes from different purifications. This has been clarified in the revised figure legend:

“E) Rates for the s1 – to – non-s1 and non-s1 – to – s1 transitions determined using ebFRET, performed as previously described³⁸, and fitting the survival plots to a double exponential. (n = 3 biological replicates with separate ebFRET analyses, contributing to N = 289 particles; error bars represent the SD).”